# Methylation-regulated decommissioning of multimeric PP2A complexes

Cheng-Guo Wu[1,2], Aiping Zheng [1,3], Li Jiang[1,4], Michael Rowse[1], Vitali Stanevich[1,5], Hui Chen[1], Yitong Li[1], Kenneth A. Satyshur[1], Benjamin Johnson[1], Ting-Jia Gu[1], Zuojia Liu[1] & Yongna Xing[1,2,5]

Dynamic assembly/disassembly of signaling complexes are crucial for cellular functions. Specialized latency and activation chaperones control the biogenesis of protein phosphatase 2A (PP2A) holoenzymes that contain a common scaffold and catalytic subunits and a variable regulatory subunit. Here we show that the butterfly-shaped TIPRL (TOR signaling pathway regulator) makes highly integrative multibranching contacts with the PP2A catalytic subunit, selective for the unmethylated tail and perturbing/inactivating the phosphatase active site. TIPRL also makes unusual wobble contacts with the scaffold subunit, allowing TIPRL, but not the overlapping regulatory subunits, to tolerate disease-associated PP2A mutations, resulting in reduced holoenzyme assembly and enhanced inactivation of mutant PP2A. Strikingly, TIPRL and the latency chaperone, α4, coordinate to disassemble active holoenzymes into latent PP2A, strictly controlled by methylation. Our study reveals a mechanism for methylation-responsive inactivation and holoenzyme disassembly, illustrating the complexity of regulation/signaling, dynamic complex disassembly, and disease mutations in cancer and intellectual disability.

[1] McArdle Laboratory for Cancer Research, Department of Oncology, University of Wisconsin at Madison, School of Medicine and Public Health, Madison, WI 53705, USA. [2] Biophysics Program, University of Wisconsin at Madison, Madison, WI 53706, USA. [3]Present address: Department of Microbiology & Molecular Genetics, University of Pittsburgh, Pittsburgh, PA 15219, USA. [4]Present address: Harbin Veterinary Research Institute, Chinese Academy of Agricultural Sciences (CAAS), Harbin, 150001 PR, China. [5]Present address: Johnson & Johnson, Melvin, PA 19355, USA. Cheng-Guo Wu and Aiping Zheng contributed equally to this work. Correspondence and requests for materials should be addressed to Y.X. (email: xing@oncology.wisc.edu)

Besides protein folding and complex assembly, disassembly of signaling complexes are also crucial for broad aspects of cellular processes[1–3], such as Hsp70-mediated disassembly/uncoating of clathrin cages from vesicles during endocytosis[4], and Hsp90 and its co-chaperone p23-mediated disassembly of transcriptional regulatory complexes in response to signaling changes[5]. Protein phosphatase 2A (PP2A), together with the closely related PP1, contributes >90% of the phosphatase activity inside cells. The broad substrate specificity and cellular functions of PP2A are controlled by formation of diverse heterotrimeric holoenzymes; each contains a common core enzyme formed by the scaffold (A) and catalytic (C or PP2Ac) subunits and a third variable regulatory subunit from four major families (B/B55/PR55, B'/B56/PR61, B"/PR72, and B"'/Striatin)[6–10]. PP2A-specific latency and activation chaperones, α4 and PTPA (PP2A phosphatase activator)[11,12], and methylation enzyme, LCMT-1 (leucine carboxylmethyltransferase 1)[13], are involved in the biogenesis of diverse PP2A holoenzymes. PP2A plays an important role in cell cycle progression, growth control, cell death and survival, DNA damage response, and cytoskeleton dynamics[7,8] and has important implications for devastating human diseases, including multiple types of cancer and Alzheimer's disease[14–17]. Likely due to PP2A's complex compositions, functions, and regulations, specialized chaperones are required and co-evolved with PP2A to specifically control the assembly and disassembly of PP2A holoenzymes. PP2A and its latency/activation chaperones, α4/PTPA, and methyltransferase LCMT-1, are all highly conserved from yeast to mammal[11–13].

The complex molecular processes of PP2A holoenzyme biogenesis are coordinated by α4/PTPA-controlled global conformational switches of PP2Ac, which coherently control PP2Ac's sequential interactions with its regulatory proteins/enzymes and subunits during holoenzyme biogenesis: (1) α4 stabilizes the partially folded, latent PP2Ac and excludes A subunit binding[12]; (2) PTPA stabilizes an active protein fold, catalyzes maturation of the active site/chelation of authentic catalytic metal ions, and facilitates A subunit binding and core enzyme formation[11]; (3) PTPA and A subunit concertedly enhance carboxyl-methylation of PP2Ac by PP2A-specific methyltransferase (LCMT-1)[13,18], which is reversibly controlled by PP2A-specific methylesterase 1 (PME-1)[19–21]; and (4) regulatory subunits bind in a mutually exclusive manner to the core enzyme to form diverse holoenzymes. The holoenzyme biogenesis pathway is critical for broad cellular functions of PP2A holoenzymes as α4, PTPA, and LCMT-1 are essential for cell survival, cell cycle progression, diverse cellular signaling, stress responses, and drug sensitivity[22–26], similar to PP2A itself.

Tight control of cellular PP2A holoenzymes is crucial for normal physiology. Diverse PP2A mutations have been suggested in developmental/intellectual disorders[27] and cancer[28,29], the mechanism for many of which remains to be characterized. Besides tight control of PP2A holoenzyme biogenesis, conversion of PP2A holoenzymes to latent PP2A was also suggested during DNA damage and stress responses[24]. Disassembly of PP2A holoenzymes is highly counterintuitive, however, as intersubunit interactions in PP2A holoenzymes were shown to be in the low nanomolar range[30]. A molecular process governing holoenzyme disassembly has never been demonstrated. The global changes of PP2Ac, associated with α4-binding near the active site that disrupts the scaffold subunit binding site on the opposite side, suggest a potential role for α4 in disassembly of PP2A holoenzymes[12]. The binding of α4, however, requires a biological trigger to perturb the phosphatase active site. TIPRL (target of rapamycin (TOR) signaling pathway regulator) is an important PP2A inhibitory protein[26] and forms a complex with PP2Ac together with α4[31], excluding both scaffold and regulatory subunits[32,33]. It

plays an important role in mammalian TOR signaling[25] and DNA damage response[26] and is upregulated in hepatocellular carcinoma, protecting cancer cells from apoptotic cell death[34].

To gain insights into the molecular mechanism of TIPRL, here we elucidate the high-resolution crystal structures of TIPRL in isolation and in complex with the PP2A core enzyme. Structure-guided studies reveal that TIPRL inhibits PP2A by perturbing the phosphatase active site, and together with α4, exhibits a robust activity in disassembly of active PP2A holoenzymes in a highly sensitive methylation-responsive manner. TIPRL forms unusual wobble contacts with the PP2A scaffold subunit that underlie enhanced holoenzyme disassembly and inhibition of holoenzyme assembly of PP2A mutants that are linked to cancer and cause intellectual disability. Our studies underlie a PP2A regulation/signaling loop of holoenzyme biogenesis and disassembly and provide general insights into dynamic signaling complex disassembly regulated by covalent modifications.

## Results

**Overall structures of TIPRL and its complex with PP2A.** We determined the crystal structures of mouse TIPRL and its complex with the PP2A core enzyme at 2.7 and 3.8 Å, respectively (Table 1). TIPRL possesses a unique butterfly-shaped protein fold with two extended layers of β-sheets that pack closely and in parallel at the center. Several helical wings (HWs) arch outward to give an overall butterfly-shaped structure (Supplementary Fig. 1a). TIPRL structure is very similar between mouse and human (Supplementary Fig. 1a)[35] and is likely conserved across species due to a high level of sequence similarity (Supplementary Fig. 1b). Particularly, the concave surface formed by the more extended second β-sheet and HWs is most conserved (Supplementary Fig. 1b).

Surprisingly, TIPRL directly contacts the PP2A scaffold subunit by sitting with the side edge of its β-sheets on the top ridge of the N-terminal HEAT (Hungtinton-elongation factor-A subunit-TOR) repeats (Fig. 1), overlapping with the binding sites of PP2A regulatory subunits with known holoenzyme structures[36–39]. The conserved concave surface of TIPRL cuddles the highly conserved C-terminal peptide motif of PP2Ac (PP2A tail), and HW2 and HW3 arch toward and hug PP2Ac at two separate binding sites near the phosphatase active site (Fig. 1). While the PP2A-TIPRL complex has dimensions similar to PP2A holoenzymes[36–39] (90 Å in length and width and 80 Å in height), TIPRL makes more extensive contacts with the catalytic subunit than any known regulatory subunits. Moreover, the complex exhibits significant structural dynamics as reflected by the structural diversity of its four asymmetric copies in the crystal (Fig. 1), with the root-mean-square-deviation (RMSD) between different copies up to 2.6 Å, revealing dynamic structural fluctuation in TIPRL–PP2A interactions. As detailed later, TIPRL is markedly different from PP2A regulatory subunits in that it binds specifically to the demethylated PP2A tail and perturbs the conformation and chelation of catalytic metal ions at the active site, resulting in phosphatase inactivation.

**Multifaceted roles of TIPRL in suppressing PP2A activity.** TIPRL makes highly integrative tripartite contacts with PP2Ac, including interactions at the PP2A tail (Fig. 2a, b) and near the phosphatase active site (Fig. 2a, c, d). The highly conserved protein groove of TIPRL (Supplementary Fig. 1b) is largely positively charged (R144/173/184/200 and K171) with interspersed hydrophobic patches (T138, L141, F180, and I147) that mediate a specific interaction with the similarly conserved C-terminal peptide motif "$D_{306}Y_{307}F_{308}L_{309}$" in the PP2Ac tail (Fig. 2b). The carboxylate group of L309—the last residue of

**Table 1 Crystallographic data collection and refinement statistics**

|  | Se-Met TIPRL (12-259 Δ94-103) | Se-Met PP2A-TIPRL (12-259) |
|---|---|---|
| Data collection |  |  |
| Space group | P6₃ | P3₂ |
| Cell dimensions (Å) |  |  |
| $a, b, c$ (Å) | 63.649, 63.649, 102.079 | 150.983, 150.983, 285.498 |
| $\alpha, \beta, \gamma$ (°) | 90.00, 90.00, 120.00 | 90, 90, 120 |
| Resolution (Å) | 49.49 –2.72 (2.78-2.72) | 50-3.8 (3.88-3.80) |
| Rmerge (%) | 12.8 (69.2) | 87.5[a] |
| $I/\sigma I$ | 31.94 (1.39) | 11.3 (1)[b] |
| Completeness (%) | 97.3 (45.9) | 100 (100) |
| Redundancy | 10.5 (5.9) | 11.7 (11.8) |
| Refinement |  |  |
| Resolution (Å) | 49.42-2.72 | 49.42-3.8 |
| No. of reflections | 12,335 | 143,300 |
| $R$-factor (%) | 20.8 | 19.7 |
| $R$-free (%) | 27.3 | 24.6 |
| No. of atoms |  |  |
| Protein | 1773 | 34,732 |
| Water | 35 | 0 |
| Average $B$-factors (Å²) |  |  |
| Protein | 86.47 | 167.47 |
| Water | 62.89 | NA |
| R.m.s. deviations |  |  |
| Bond lengths (Å) | 0.011 | 0.004 |
| Bond angles (°) | 1.4 | 1.05 |
| Ramachandran plot |  |  |
| Favored (%) | 94.23 | 92.57 |
| Allowed (%) | 5.77 | 7.0 |
| Outliers (%) | 0 | 0.4 |

NA not applicable. Values in parentheses are for the highest-resolution shell
[a] This dataset is a result of merging and scaling of three datasets
[b] The value of CC₁/₂ is 0.105. The resolution cutoff of 3.80 Å is estimated based on CC1/2 fit of 0.30

PP2Ac—is nudged closely to the positively changed side chain of R200 in TIPRL, exhibiting well-defined salt bridge and H-bond interactions (Fig. 2b). This interaction elegantly defines TIPRL's selectivity for demethylated PP2Ac. The binding affinity between TIPRL and the PP2A core enzyme was measured at ~0.31 μM (Fig. 2f, left and right panels), and methylation of PP2Ac rendered this interaction almost undetectable (Fig. 2f, middle panel). Moreover, the interaction between PP2A and TIPRL was abolished by alterations of the PP2Ac tail, the Y307E mutation or deletion of residue L309, or by changes to TIPRL residues at the interface with PP2Ac tail (Fig. 3a–c).

Near the PP2A active site, HW3 of TIPRL makes well-defined hydrophobic and H-bond interactions to the β12–β13 and β3–α3 loops (Fig. 2c). HW2 directly contacts the helix switch of PP2Ac (Fig. 2d). The helix switch was previously shown to be relaxed in the α4–PP2Ac complex, enabling global changes in PP2Ac that perturb the scaffold subunit binding site opposite to the phosphatase active site[12]. Mutation of TIPRL residues at either interface abolished or weakened interactions between TIPRL and the PP2A core enzyme (Fig. 3d) without obvious alteration of the thermal stability of TIRPL (Supplementary Fig. 2). Both β12–β13 loop and helix switch at the interface with TIPRL are different from those observed in active PP2A bound to okadaic acid (OA), a highly potent PP2A inhibitor and tumor-inducing toxin[40] (Fig. 2c, d). Consistently, albeit their binding sites do not overlap (Supplementary Fig. 3a), OA blocks TIPRL binding (Supplementary Fig. 3b), suggesting that OA stabilizes the PP2A active site in a conformation unfavorable for TIPRL-binding (Fig. 2a, c, d). TIPRL binding near the PP2A active site distorted PP2A active site loops and metal chelation residues, thus perturbing proper chelation of the two catalytic metal ions, with a misplaced metal ion left several angstroms from the active site that would not confer any enzymatic activity (Fig. 2e). Reciprocally, stabilization of the PP2A active site by an excess amount of free Mn²⁺ weakened TIPRL binding (Fig. 3e). The loss of precise catalytic metal chelation upon TIPRL's distortion of the active site correlated with a loss of phosphatase activity. In accordance with these data, TIPRL inhibited the phosphatase activity of the PP2A core enzyme with an IC₅₀ of ~0.3 μM in the absence of free Mn²⁺ (Fig. 2g), comparable to the measured binding affinity between TIPRL and PP2A (Fig. 2f, left and right panels). Upon carboxylmethylation of the PP2A core enzyme, the IC₅₀ of TIPRL was increased by ~30-fold (Fig. 2g), further supporting the notion that methylation serves as a signal switch that prevents TIPRL from interacting with PP2A or inactivating the PP2A active site.

Structural overlay shows that TIPRL overlaps with PTPA and LCMT-1 for binding near the PP2A active site and thus sterically hinders the binding of PTPA and LCMT-1[13,41] (Fig. 4a), both of which are also known to directly contact the PP2A tail[13,42], similar to TIPRL. Thus TIPRL competes with PTPA and LCMT-1 for interactions with the PP2A core enzyme (Fig. 4b, c). Besides inhibition of the PP2A core enzyme, TIPRL also inhibited the unmethylated holoenzyme from the B'/B56/PR61 family but not PP2A-B"/PR70 holoenzyme or methylated holoenzymes (Fig. 4d, e). Thus, TIPRL might play a negative multifaceted role in holoenzyme biogenesis and function, and methylation might be an important mechanism to prevent TIPRL's perturbation of holoenzyme integrity (Fig. 4f).

**Wobble TIPRL contacts and role in PP2A disease mutations.**
TIPRL sits on the top ridge of HEAT repeats 2–6 of the PP2A scaffold subunit via the side edge of TIPRL's elongated double-

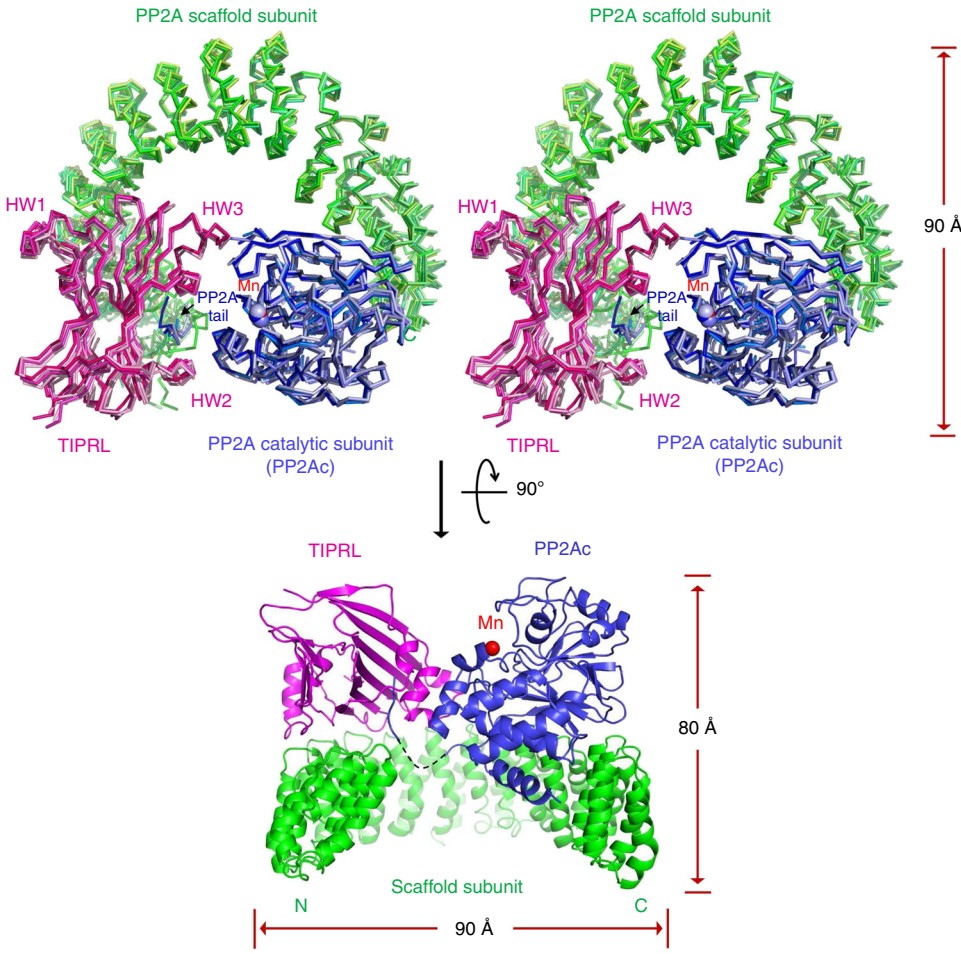

**Fig. 1** Overall structure of the PP2A–TIPRL complex. Top: Stereo view of overlaid crystal structures of the four copies of the PP2A–TIPRL complex in each asymmetric unit in ribbon. Bottom: the perpendicular view of one copy of the PP2A–TIPRL complex in cartoon. The PP2A scaffold subunit, catalytic subunit, and TIPRL are colored in green, blue, and magenta, respectively. Different copies of each protein are colored in related colors. Manganese ions are shown in blue (top) and red (bottom) spheres. The disordered region of catalytic subunit (residue 295–303) is indicated by dashed line (bottom)

layer β-sheets (Fig. 5a). These contacts overlap the binding sites of three families of PP2A regulatory subunits with known holoenzyme structures (Supplementary Fig. 4). The contacts between TIPRL and PP2A scaffold subunit are in highly diverse modes as exemplified by overlaying the four copies of the complex in the crystallographic asymmetric unit by TIPRL (Fig. 5a, b). Several H-bond, salt bridge, and π–π interactions are found varied significantly among different copies of the complex, surrounding the common hydrophobic contacts at the center of the interface (Fig. 5a, upper right, circled, and Supplementary Fig. 5). The tolerance of different modes of interactions at the periphery of the A-TIPRL interface resembles the tolerance of different nucleotides at the third wobble position of codons. We thus refer to these contacts as "wobble contacts". The A-TIRPL wobble contacts might be in part contributed by the highly dynamic nature of the scaffold subunit (Supplementary Movie 1). Consistent with this notion, the RMSD of the scaffold subunit between different copies of the complex is up to 2.9 Å. In contrast, the RMSD of TIPRL between different copies is much smaller, 0.6–0.7 Å. Intriguingly, TIPRL tolerates a broad range of scaffold subunit conformations observed in diverse PP2A holoenzymes[9,36–39] (Fig. 5b). By contrast, different functional holoenzymes tend to adopt highly distinct scaffold subunit conformations specific for each regulatory subunit[36–39]. Direct contacts between TIPRL and the PP2A scaffold subunit had not been observed previously. Consistent with our structural observations, in the absence of the

scaffold subunit, the interaction between PP2Ac and TIPRL is weakened from 0.3 μM (Fig. 2f, left and right panels) to ~0.6 μM (Supplementary Fig. 6a), despite that only a very weak interaction could be detected between the scaffold subunit and TIPRL (~10 μM, Supplementary Fig. 6b). TIPRL mutations at the interface with the scaffold subunit weakened the interaction between TIPRL and PP2A both in vitro and in mammalian cells (Fig. 5c, d), underlying a finding that these contacts were important for PP2A–TIPRL interactions. Since PP2Ac is predominantly associated with the scaffold subunit in cells, disrupting the interactions between TIPRL and the scaffold subunit abolished the detection of interactions between TIPRL and PP2Ac (Fig. 5d).

Dynamic wobble contacts between TIPRL and the scaffold subunit may underlie important mechanisms by which PP2A disease mutations interfere with holoenzyme assembly. Somatic mutations of PP2A are frequently found in cancer[28,29]. Moreover, de novo mutations in PP2A have been linked to severe intellectual disability observed in children[27]. Several of these mutations (P179R, R183Q, R183W, S256F, and R258H) occur in the PP2A scaffold subunit at the interface to TIPRL and the PP2A regulatory subunits (Supplementary Fig. 7)[36–39]. Strikingly, we found that these disease-associated mutations did not interfere with or even slightly enhance TIPRL binding but significantly weakened the binding of regulatory subunits, as shown for B'γ1 (Fig. 5e, f). Although the R258H mutation retained B'γ1 binding, the ability of TIPRL to compete with B'γ1 was significantly

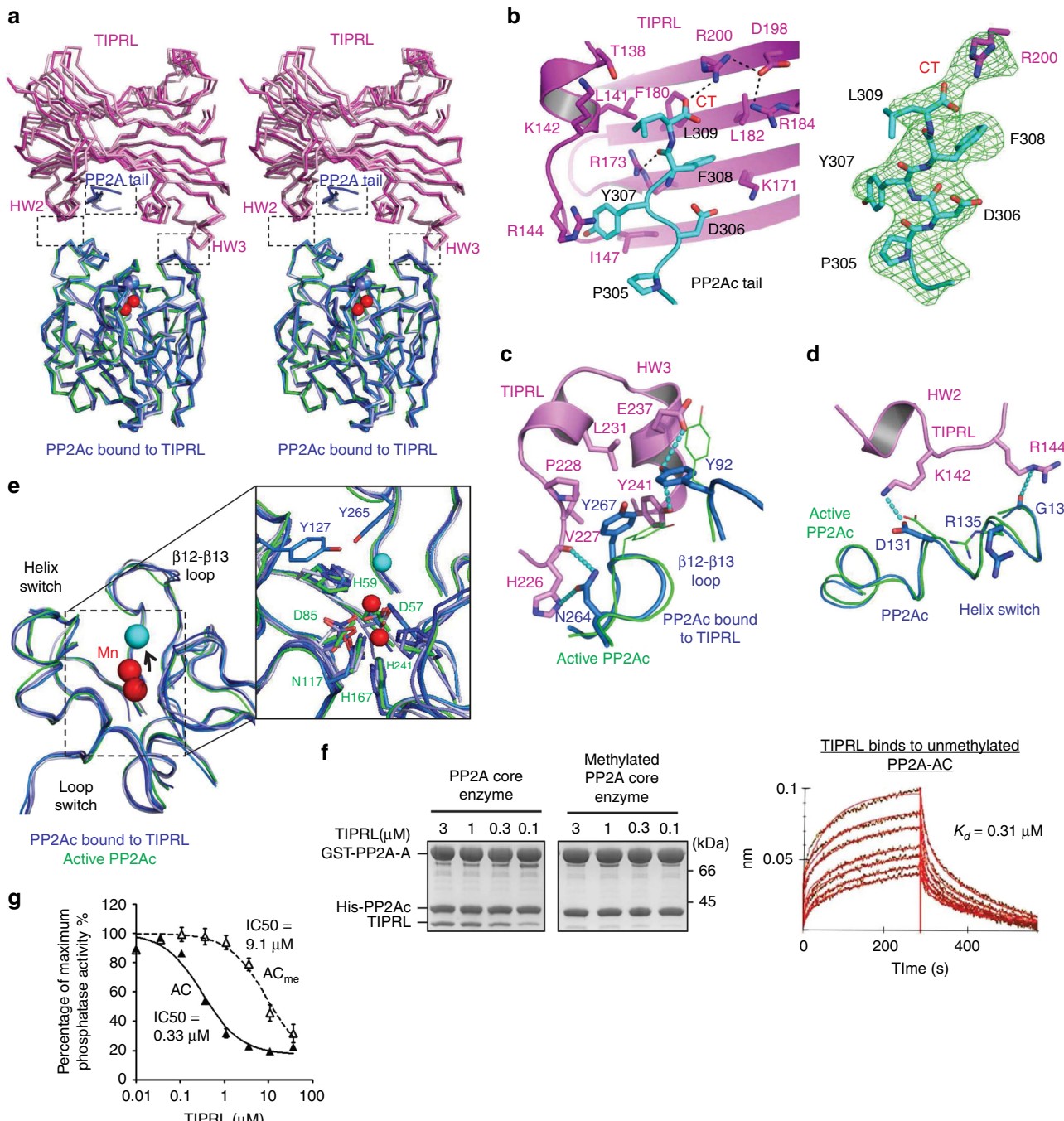

**Fig. 2** Interactions between PP2Ac and TIPRL. **a** An overview of the interactions between PP2Ac (blue) and TIPRL (magenta) shown in ribbon with dashed boxes highlighting three separate interaction interfaces. Four copies of the complex in the asymmetric unit are overlaid. Manganese ions in active and TIPRL-bound PP2Ac are in red and blue spheres, respectively. **b** A close-up view of the interaction between PP2Ac tail (cyan) and TIPRL (left). The $F_O$-$F_C$ omit map for PP2Ac tail and the side chain of R200 of TIPRL is contoured at 3.0σ in green mesh (right). CT stands for C-terminus of PP2Ac. H-bond interactions are shown in black dashed lines. **c**, **d** Interaction interfaces of HW3 and HW2 of TIPRL with the structural elements near the active site of PP2Ac shown in cartoon, overlaid with active PP2Ac as in **a**. The color scheme is the same as in **a**. Residues in the TIPRL–PP2Ac complex are shown in sticks and those in active PP2Ac in lines. H-bond interactions are shown in cyan dashed lines. **e** Structural comparison of active sites of the active (green) vs TIPRL-bound (blue) PP2Ac, with manganese ions in red and cyan spheres, respectively. The right box shows the close-up view of the chelation of manganese for active and TIPRL-bound PP2Ac. Key residues involving in chelation are labeled in green and blue for active and TIPRL-bound PP2Ac, respectively. **f** Pull-down of titrated concentrations of TIPRL by non-methylated (left) vs methylated (middle) PP2A core enzyme containing the GST-tagged scaffold subunit (PP2A-A). The bound proteins were examined on SDS-PAGE. Association and dissociation curves of TIPRL to BLI surface activated by GST-tagged unmethylated PP2A core enzyme with calculated binding affinity (right). **g** Phosphatase activity of PP2A core enzyme in the presence and absence of increasing concentrations of TIPRL, normalized in percentage to the activity in the absence of TIPRL. Experiments were performed in triplicate and repeated three times. Data are shown as mean and standard error of the mean (s.e.m.)

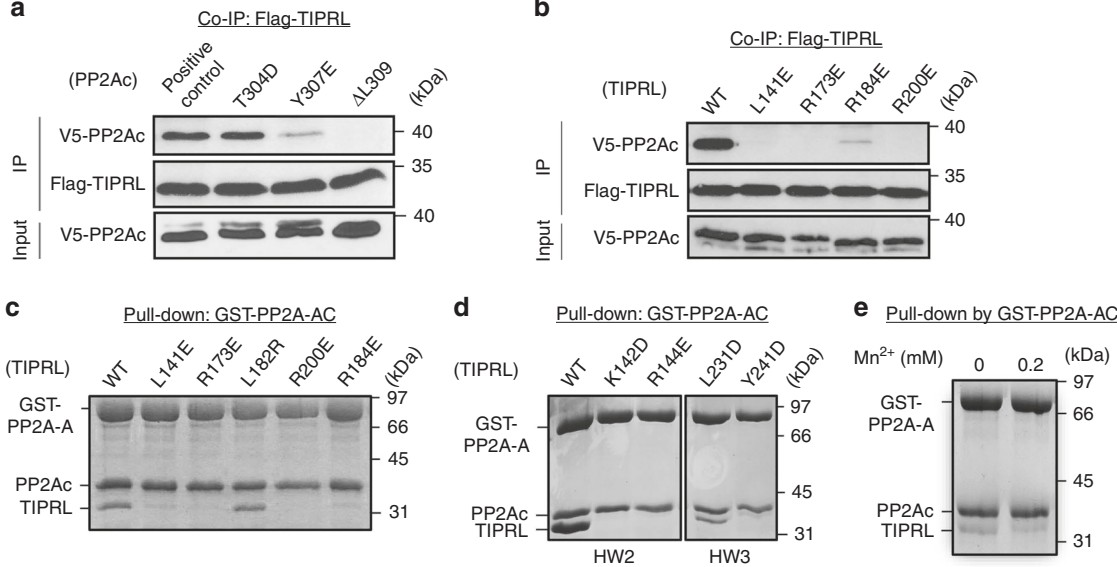

**Fig. 3** Mutagenesis and biochemical characterization of PP2Ac–TIPRL interfaces and their modes of interaction. **a** Co-IP of WT and mutant V5-tagged PP2Ac bearing mutations in PP2Ac tail by Flag-tagged TIPRL. **b** Co-IP of V5-tagged PP2Ac by WT and mutant Flag-tagged TIPRL bearing mutations at the interface to PP2Ac tail shown in Fig. 2b. **c** Pulldown of WT and mutant TIPRL bearing mutations on residues involving in interaction with PP2Ac tail by PP2A core enzyme containing GST-PP2A-A. **d** Pulldown of WT and mutant TIPRL bearing mutations at the interface to structural elements near the PP2A active site by GST-PP2A core enzyme as in **c**. **e** Pulldown of TIPRL by GST-PP2A core enzyme in the presence or absence of $Mn^{2+}$. For **c**–**e**, proteins remained bound to GS4B resin were examined on SDS-PAGE

enhanced by this mutation (Fig. 5g). This result is in line with a recent observation that these mutations result in a higher level of TIPRL associated with cellular PP2A in cancer[43]. These mutations are expected to perturb either the direct contacts or the local folding at the interface with TIPRL/regulatory subunits (Supplementary Fig. 7). The fact that they specifically perturbed regulatory subunit binding but not TIPRL corroborated strongly with the dynamic wobble interactions predicted to sustain TIPRL–PP2A binding (Fig. 5a, b). In sum, these structural and biochemical insights implicate TIPRL–PP2A in dysregulated cell signaling underpinning PP2A-linked cancers and neurological disorders.

**Holoenzyme disassembly by α4 and TIPRL.** How active PP2A complexes are inactivated and disassembled in the cell is unknown. Our previous observation of global changes of PP2Ac associated with α4 binding suggested a potential mechanism for disassembly of PP2A holoenzymes[12] but required a "trigger" protein or cellular factor that disrupts or attacks the phosphatase active site. Several structural features of TIPRL-bound PP2Ac suggested that TIPRL is a strong candidate for such an activity. First, TIPRL binding significantly altered the phosphatase active site that perturbed chelation of catalytic metal ions (Figs. 2e and 6a). Second, some of the central β-strands of PP2Ac exhibit distinctly different conformations and are even partially disordered in the TIPRL-bound complexes (Fig. 6a, upper right). The RMSD of PP2Ac between different copies of the TIPRL-bound complexes is up to 1.4 Å, much higher than that of TIPRL. Third, close contacts between PP2Ac and the scaffold subunit (with a binding affinity of 5 nM in the active enzyme[30]) exhibited significant wobble in the TIPRL-bound complexes (Fig. 6a, lower right). Finally, B factors of the TIPRL-bound PP2Ac are globally higher than the active core enzyme or higher near the active site (Supplementary Fig. 8), suggesting a higher structural flexibility when bound to TIPRL. These protein structural fluctuations observed in the crystal structures are expected to be more dynamic in solution. Thus TIPRL binding might facilitate α4

binding near the PP2A active site to trigger more drastic global changes and collectively disassemble active PP2A holoenzymes.

The notion of active holoenzyme disassembly was perceived to be counterintuitive, considering that intersubunit interactions within the PP2A holoenzymes are of very high affinity (at the low nanomolar range)[30]. However, that perturbing the PP2A active site affected the global conformation of PP2Ac and its capacity to bind scaffold subunit[12] prompted us to test whether active disassembly occurs. To this end, the active PP2A-B'γ1 holoenzyme was co-incubated with a near stoichiometric amount of glutathione S-transferase (GST)-α4 and TIPRL at 37 °C. PP2Ac was gradually extracted from the holoenzyme into the form of a trimeric latent PP2Ac–α4–TIPRL complex (Fig. 6b, c). This effect was similar to co-incubation of the holoenzyme with α4 and pyrophosphate (PPi), conditions known to chemically evict the catalytic metal ions[12]. Moreover, α4 alone barely affected the stable active holoenzyme (Fig. 6b, c), demonstrating TIPRL's capacity to trigger holoenzyme disassembly and supporting our hypothesis. Consistent with the role of methylation in protecting holoenzyme integrity, α4/TIPRL failed to extract PP2Ac from the methylated holoenzyme (Fig. 6b, c).

In the presence of free excess $Mn^{2+}$, which stabilizes the phosphatase active site and weakens TIPRL binding (Fig. 3e), neither α4 or TIPRL alone could inactivate the core enzyme (Supplementary Fig. 9, left). In contrast, the two proteins together rapidly inactivated the complex. Inductively coupled plasma mass spectrometry (ICP-MS) detected a drastic loss of catalytic metal ions in PP2A after incubation with α4/TIPRL together, and a marginal loss with TIPRL alone (Supplementary Fig. 9, right), demonstrating the robust, combined force of α4/TIRPL in attacking the phosphatase active site to dislodge catalytic metal ions and promote disassembly of PP2A complexes. The α4-bound latent PP2Ac is essential for the biogenesis of diverse PP2A holoenzymes (Fig. 4f)[12]. Thus we suggest that α4/TIRPL-mediated disassembly of PP2A holoenzymes serves as a general mechanism for recycling of PP2Ac, thereby allowing it to complex with different regulatory subunits to form diverse

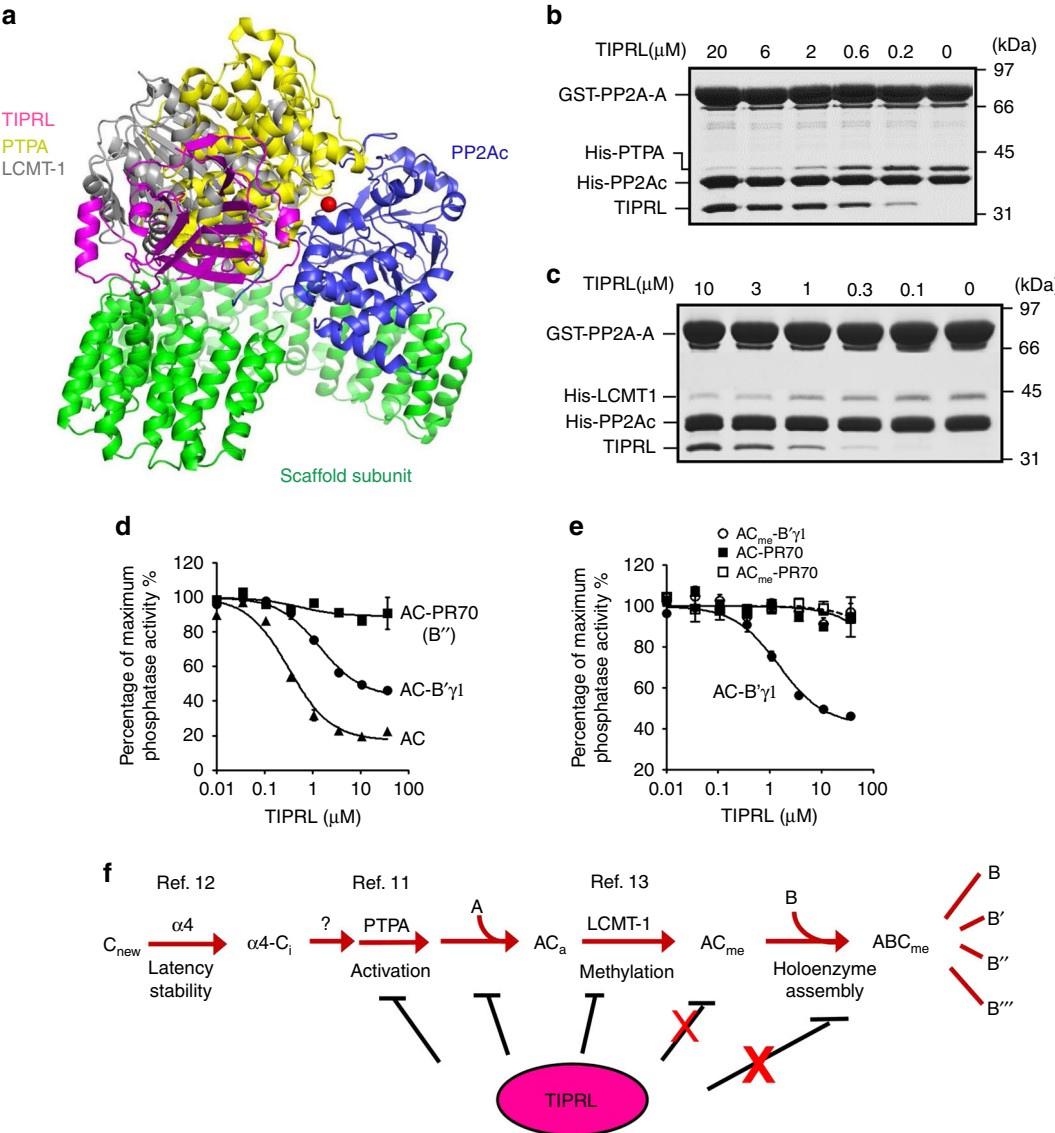

**Fig. 4** Effects of TIPRL on the molecular events en route of holoenzyme biogenesis. **a** Overlay of crystal structures of PP2A-TIPRL (PDB code: 5W0W), PP2Ac-PTPA (PDB code: 4LAC), and PP2Ac-LCMT-1 (PDB code: 4I5L) by catalytic subunit illustrates the overlapping binding of TIPRL, PTPA, and LCMT-1 to the PP2A active site. The structures are shown in cartoon and colored as in Fig. 1 for the PP2A–TIPRL complex and colored yellow and gray for PTPA and LCMT-1, respectively. **b** Pulldown of PTPA in the presence and absence of titrated concentrations of TIPRL by immobilized GST-PP2A core enzyme. **c** Pulldown of LCMT-1 in the presence and absence of titrated concentrations of TIPRL by immobilized GST-PP2A core enzyme. For **b**, **c**, proteins remained bound to GS4B resin were examined on SDS-PAGE. **d** Phosphatase activity of PP2A core enzyme and PP2A-B′γ1 and PP2A-PR70 holoenzymes in the presence and absence of increasing concentrations of TIPRL. Experiments were performed in triplicate and repeated three times. Data are shown as mean ± s.e.m. **e** Phosphatase activity of methylated and non-methylated PP2A-B′γ1 and PP2A-PR70 holoenzymes in the presence and absence of increasing concentrations of TIPRL. Data are shown as mean ± s.e.m. **f** Illustration of PP2A holoenzyme biogenesis pathway and the inhibitory effects of TIPRL on the molecular events en route of holoenzyme biogenesis. The inhibitory effects of TIPRL are weakened by PP2A methylation

PP2A holoenzymes for spatiotemporal control of PP2A activity in response to changes of cellular signaling.

## Discussion

Our work elucidated striking structural features of TIPRL as a dynamic PP2A inactivator and driver of holoenzyme disassembly in response to demethylation. These findings reveal a dynamic aspect of PP2A regulation and provide the structural mechanism for how stable PP2A holoenzymes undergo dynamic disassembly and recycling. Together with the pathway for controlling PP2A holoenzyme biogenesis (Fig. 4f), α4/TIPRL-mediated holoenzyme disassembly and recycling of PP2Ac provide a feedback

mechanism for precise regulation of PP2A holoenzyme turnover during cellular signaling, without perturbing the cellular level of PP2Ac. These results explain how PP2Ac levels remain constant during cell cycle in mammalian cells[7,8].

In this feedback loop, methylation of PP2Ac tail serves as a major signaling switch for controlling holoenzyme stability and disassembly. TIPRL's attack of the PP2A active site requires that the PP2Ac tail is demethylated (Fig. 2), which reconciles a long-standing controversy regarding the role of carboxylmethylation on PP2A holoenzyme assembly in vitro and in vivo. Methylation was found to be essential for holoenzyme assembly in cells[44,45] but not in vitro[37,40]. It is also intriguing to consider that periodic demethylation of PP2A occurs during cell cycle[46], so that PP2A

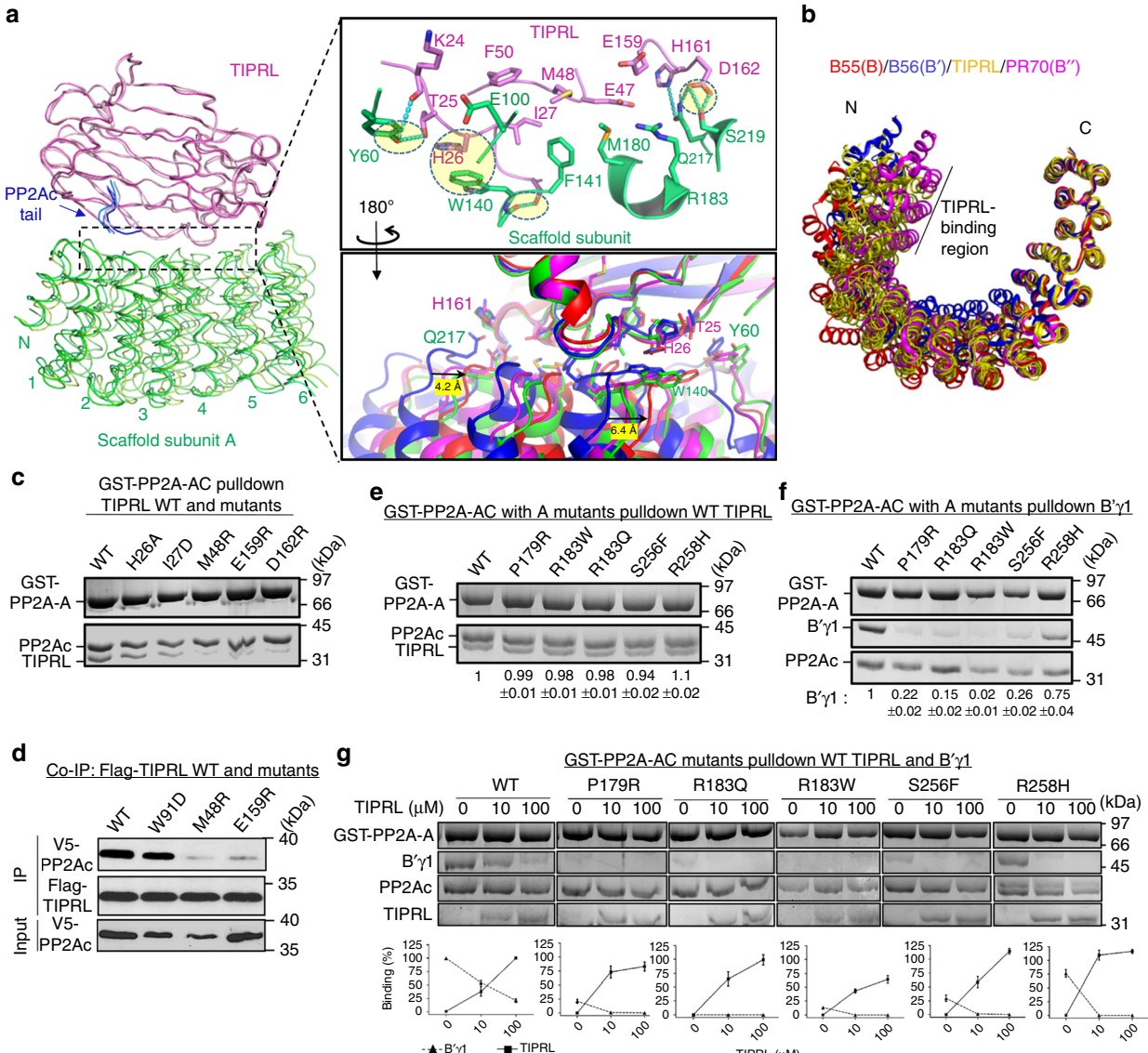

**Fig. 5** Wobble contacts of TIPRL to PP2A scaffold subunit and insights into PP2A disease mutations. **a** An overview (left) and close-up views (right) of interactions between TIPRL and the scaffold A subunit. The left overview shows the overlay of the four asymmetric copies of the complex in ribbon with the same color scheme as the upper panel of Fig. 1. The N-terminal six HEAT repeats of the scaffold subunit were shown. The upper right close-up view shows a representative interface in cartoon and colored as in overview. Key residues at the interface are shown in sticks. The wobble contacts are highlighted in circles. The bottom right close-up view shows the overlaid interfaces of all four asymmetric copies of the complex colored in blue, cyan, green, and red. The blue and red copies differ the most among the four copies. **b** Overlay of PP2A scaffold subunit in four asymmetric copies of the TIPRL-bound complex (yellow) and in different holoenzymes, PP2A-B55α (red, PDB code: 3DW8), PP2A-B'γ1 (blue, PDB code: 2NPP), and PP2A-PR70 (magenta, PDB code: 4I5L). The HEAT repeats that are involved in interaction with TIPRL are indicated by the length of a black line. **c** Pulldown of wild-type (WT) and mutant TIPRL by PP2A core enzyme containing GST-A. Proteins associated with GS4B resin were examined on SDS-PAGE. **d** Co-immunoprecipitation (Co-IP) of WT and mutant Flag-tagged TIPRL with V5-tagged PP2Ac. Pulldown of **e** TIPRL, **f** B'γ1, and **g** B'γ1 with titrated concentrations of TIPRL, by PP2A core enzyme containing WT or mutant GST-A. The mutations in the scaffold A subunit were found in cancer and intellectually disabled patients. Proteins associated with GS4B resin were examined on SDS-PAGE. The relative amount of TIPRL (**e**, **g**) and B'γ1 (**f**, **g**) associated with PP2A core enzyme were normalized to PP2A-A and then quantified as the ratio to or percentage of the normalized amount of proteins bound to the WT PP2A core enzyme in the absence of competitor. The quantified results were presented as mean ± s.e.m. and shown at the bottom of the gel (**e**, **f**, **g**) and plotted as concentration-dependent binding curves (**g**)

holoenzymes may undergo cell cycle-dependent disassembly by α4 and TIPRL and their phosphatase activity may fluctuate similar to cell cycle-dependent kinases but in opposite phases. Moreover, α4/TIPRL-mediated recycling might also contribute to downregulation of PP2A holoenzymes during stress or DNA damage responses[24]. TIPRL was found to suppress PP2A activity in DNA damage-induced ATM/ATR signaling[26]. Taken together, unique features of α4/TIPRL structure provide a novel platform

to investigate dynamic control of PP2A holoenzymes in diverse cellular signaling and cellular processes.

Furthermore, the butterfly-shaped TIPRL provides a remarkable multibranching machinery capable of forming several highly integrative contacts with PP2Ac and the scaffold subunit. Complex interactions are likely to explain TIPRL's multifaceted roles in regulating PP2A, including inactivation of the phosphatase active site, responding to demethylation, and inhibiting diverse

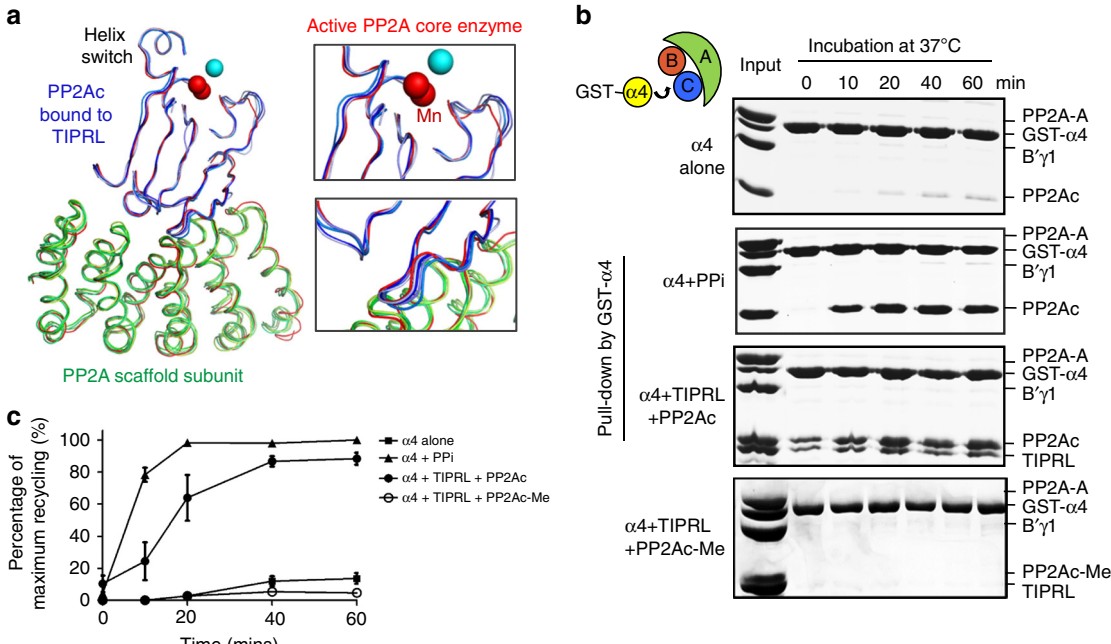

**Fig. 6** TIPRL-mediated structural attacks on PP2Ac and α4/TIPRL-mediated PP2A holoenzyme disassembly. **a** Overlaid structures of the active PP2A core enzyme (red) and the four asymmetric copies of the TIPRL-bound core enzyme (colored as in Fig. 1), with close-up views on the active site (upper right) and the A-PP2Ac interface (lower right). Manganese ions bound to active and TIPRL-bound PP2Ac are in red and blue spheres, respectively. **b** Co-incubation of unmethylated or methylated PP2A-B′γ1 holoenzyme with GST-α4 in the presence or absence of PPi or TIPRL at 37 °C, followed by pulldown via GS4B after the indicated incubation times. The bound proteins were examined on SDS-PAGE. **c** Time-dependent recycling of PP2Ac with the α4-bound PP2Ac at different time points were normalized to GST-α4 and then quantified as percentage of that at 60 min of "α4 + PPi"-mediated recycling. The quantified mean ± s.e.m. were then plotted

molecular events relevant to holoenzyme biogenesis. TIPRL's wobble contacts with the scaffold subunit are unusual, unique among all known PP2A-interacting proteins, and suggest a mechanism wherein TIPRL can accommodate interactions with both wild-type and mutant PP2A implicated in cancer and neurological disorders. Under aberrant circumstances, TIPRL might effectively compete with overlapping regulatory subunits and thus suppress PP2A functions in these patients. While methylation serves as a protection mark for PP2A holoenzymes in a cellular environment for escaping attacks by TIPRL, reduced methylation had been associated with neurological disorders[16,17], underlying that α4/TIPRL might contribute to decommissioning of holoenzyme function under pathological conditions. Other disease-associated mutations could also affect holoenzyme integrity and render PP2A holoenzymes vulnerable. Many disease-associated PP2A mutations are found in PP2A regulatory subunits distal from interfaces with PP2Ac or scaffold subunit[27–29,47,48]. Whether these mutations add additional marks to the holoenzyme relevant to recycling is yet to be tested.

Built on our studies, we propose a model for α4 and TIPRL action in holoenzyme disassembly (Fig. 7). For the methylated holoenzyme, methylation hinders initial attack by TIPRL and the stabilized active site hinders subsequent α4 binding (Fig. 7a). For the unmethylated holoenzyme, on the other hand, TIPRL exerts an initial attack on the PP2A active site and causes partial release of catalytic metal ions. TIPRL would also partially compete with the regulatory subunit and facilitate α4 docking to the active site. Their combined force to the active site and the protein fold of PP2Ac triggers global conformational switches that expel all catalytic metal ions and scaffold subunit binding. By this way, α4/TIPRL cooperate to trigger disassembly of the holoenzyme complex (Fig. 7b). An overlay of the structures of the partial PP2Ac–α4 complex and the PP2A–TIPRL complex suggests that

there might be direct contacts between TIPRL and α4, which might create additional forces in the complex that contribute to global changes in PP2Ac (Supplementary Fig. 10). Elucidation of the structure of PP2Ac–α4–TIPRL recycling complex and molecular dynamic simulation of the recycling process will provide further structural insights into this striking molecular mechanism for protein complex disassembly.

It is important to mention that PP4 and PP6 are considered PP2A-like phosphatases; they have a much higher sequence similarity to PP2A than other members of the PPP family. Consistently, PP4 and PP6 shared several regulatory proteins with PP2A, including methylation enzyme, latency chaperone α4[12], and inhibitory protein TIPRL[26]. TIPRL was also found to suppress PP4 activity during DNA damage response[49]. To predict whether the mode of interaction between PP4/PP6 and TIPRL is similar to PP2A–TIPRL, we performed sequence alignment of PPP family serine/threonine phosphatases to examine residues involved in TIPRL interaction (Supplementary Fig. 11). All the residues in PP2A that participate in TIPRL binding are identical in PP4, and only one residue is different in PP6, suggesting that TIPRL interacts with PP4 and PP6 in a similar mode as with PP2A. Interestingly, the helix switch near the TIPRL-binding site has identical sequence in PP4 and PP6 as well, suggesting that the forces created by TIPRL binding might lead to similar structural strain in PP4 and PP6 to trigger their dissociation from holoenzymes.

Compared to disassembly of cell signaling complexes mediated by p23, Hsp90, and Hsp70[1–5], α4/TIPRL-mediated holoenzyme disassembly appears to have distinct features including: (1) no ATP molecule is needed to power the disassembly process; (2) PP2Ac dissociated from holoenzymes is stabilized in an α4-bound latent form and is thus expected to be recycled for holoenzyme biogenesis, rather than being degraded; (3) effective removal of

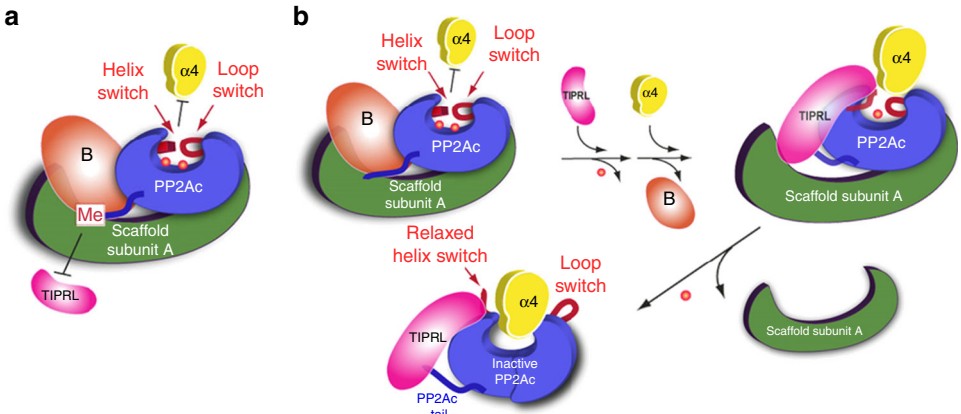

**Fig. 7** Cartoon models for α4/TIPRL-mediated PP2A holoenzyme disassembly and recycling of PP2Ac into the latent PP2Ac-α4-TIPRL complex. **a** Cartoon illustration of mechanisms protecting methylated holoenzyme from TIPRL/α4 attack. In the methylated holoenzyme, the methylated PP2Ac tail hinders TIPRL binding and thus weakens TIPRL's ability to attack the phosphatase active site. The intact active site would hinder α4 binding and subsequent global conformational changes[12]. The cartoon for the PP2A scaffold subunit is colored green, PP2Ac blue, regulatory subunit brown, TIPRL magenta, α4 yellow, and helix and loop switches of PP2Ac red. Catalytic metal ions are indicated by red dots. **b** Cartoon illustration of TIPRL/α4-mediated disassembly of unmethylated holoenzyme. Without methylation, the holoenzyme is subjected to initial attacks by TIPRL, which induces partial dislodge of catalytic metal ions and potential dissociation of the regulatory subunit, as well as partial perturbation of helix and loop switches near the PP2A active site to allow α4 docking. The combined force of α4 and TIPRL triggers global conformational switches of PP2Ac that completely release catalytic metal ions and expel both scaffold and regulatory subunits, resulting in the latent PP2Ac–α4–TIPRL complex

catalytic metal ions had not been observed for any known protein chaperones. The entire signaling feedback loop regulating PP2A holoenzyme biogenesis and recycling, however, is powered by ATP for folding latent PP2Ac into an active conformation, which might require common protein chaperones. The maturation of the active site also requires ATP. The specialized PP2A activation chaperone, PTPA, forms a combined ATP-binding pocket together with the phosphatase active site. The precisely positioned γ-phosphate of ATP modifies the metal chelation chemistry at the active site and increases the activity of $Mg^{2+}$ in PP2A activation by 10,000-fold[11]. Considering the needs of methylation and demethylation in PP2A holoenzyme biogenesis and recycling, this signaling loop is arguably powered by methylation and demethylation of PP2Ac tail. While Hsp90 uses adaptor proteins, which are often tetratricopeptide (TPR) domain-containing proteins, for interaction with client proteins for both activation and disassembly of signaling complexes[2,50], it is intriguing that α4 is a TPR domain protein and plays a role in both holoenzyme disassembly and establishing stable latency for holoenzyme biogenesis[12]. Thus further mechanistic understanding of PP2A holoenzyme biogenesis and recycling will provide distinct and general insights into signaling complex assembly and disassembly regulated by covalent modifications.

## Methods

**Molecular cloning and protein preparation**. All constructs and point mutations were generated using a standard PCR-based cloning strategy. Primers used in this study are listed in Supplementary Table 1. The construct used for crystallization of mouse TIPRL contains residues 12–259 with an internal deletion of residues 94–103 and truncation at the C-terminal region (residue 260–271). The construct used for crystallization of PP2A–TIPRL complex contains human PP2A Aα (8–589), PP2A Cα (1–309), and mouse TIPRL (12–259). Full length of wild-type (WT) or mutants of PP2A Cα, TIPRL, and PP2A Aα (8–589) were generated for biochemical experiments. TIPRL constructs were cloned into pQLinkH (Addgene) vector and overexpressed overnight at 23 °C in *Escherichia coli* BL21 (DE3) (New England Biolabs). The soluble fraction of *E. coli* cell lysates was purified over Ni-NTA (Qiagen). After removal of His6-tag by TEV protease, the protein sample was further fractioned by anion exchange chromatography (Source 15Q, GE Healthcare) and gel filtration chromatography (Superdex 200, GE Healthcare). Cloning, expression, and purification of human PP2A A (α) and C (α) subunits, and assembly of PP2A AC dimer followed similar procedures as described previously[40]. Briefly, the full-length, truncated and mutated PP2A Aα subunit were cloned into pGEX-2T vector (GE Healthcare) and overexpressed in *E. coli* strain BL21(DE3).

The soluble fraction of the *E. coli* cell lysate was purified by glutathione resin (Qiagen) and further fractionated by ion exchange Source 15Q column. Multiple expression cassettes of His8-tagged PP2A Cα subunit (1–309) were cloned into an engineered FastBac vector (Invitrogen). Bacmids of PP2A Cα were prepared from the Bac-to-Bac Baculovirus expression system (Thermo Fisher Scientific). Hi-5 suspension cells (Thermo Fisher Scientific) at a density of $1.5 \times 10^6$ cells ml$^{-1}$ were infected with recombinant PP2ACα baculovirus and shaken at 100 rpm for 48 h at 27 °C. Cells were then harvested by centrifugation at 1500 rpm for 10 min and lysed. The soluble fraction of cell lysates was purified by Ni-NTA resin to homogeneity and further fractionated by Source 15Q. The PP2A core enzyme was assembled by passing Cα through an excess amount of GST-Aα immobilized on glutathione resin. The AC dimer was released by on-column thrombin cleavage and further purified by ion-exchange chromatography to remove free scaffold subunit. Preparation of α4, B'γ1, PR70, LCMT-1, and PTPA were similar to our previous studies[11,13,36,37]. In brief, the WT, truncated and mutated α4, B'γ1, PR70, PTPA, and LCMT-1 (20–338) were cloned into pQLinkG (Addgene), pGEX-6P (GE Healthcare), pQlinkG, pET21b, and pET15b (Invitrogen) vectors, respectively. The above constructs were transformed into *E. coli* BL21 (DE3) strain and over-expressed overnight at 23 °C. The soluble fractions of the *E. coli* cell lysate were purified over GS4B resin for α4, B'γ1, and PR70, and Ni-NTA resin for PTPA and LCMT-1. Depending on the purpose of experiments, tags of purified proteins were retained or removed by TEV (pQlinkG), PreScission (pGEX-6P), or Thrombin (pET15b) proteases and further fractioned by Source 15Q column except that for LCMT-1, which was fractionated by Source 15S (GE Healthcare) column. Fractions containing pure target proteins were then collected and purified by Superdex 200.

**Protein crystallization and data collection**. Crystals of selenomethione-labeled mouse TIPRL (12-259 Δ94-103) and the PP2A–TIPRL complex, containing native TIPRL (12-259) and full-length PP2Ac and selenomethione-labeled scaffold subunit (8-589) were grown using the sitting-drop, vapor diffusion method at 18 °C by mixing 0.2 μl protein solution and 0.2 μl reservoir solution. Best crystals of mouse TIPRL were obtained 1 day after mixing 12 mg ml$^{-1}$ protein and 0.1 M Cacodylate at pH 6.5, 15% PEG1000 (v/v), and 0.2 M magnesium chloride. Crystals of the PP2A–TIPRL complex were obtained 2 days after incubation of 10–15 mg ml$^{-1}$ protein complex and 0.1 M citric acid at pH 5.2, 9% PEG 4000, and 0.2 M sodium acetate. All crystals were cryo-protected in reservoir solution supplemented with 20–25% ethylene glycol (v/v) and flash frozen in liquid nitrogen for data collection. Datasets with anomalous signal of selenium-labeled proteins were collected for 360 degrees at the Life Sciences Collaborative Access Team (LS-CAT) 21-ID-G beamline at the Argonne National Laboratory (APS) and processed using HKL2000[51].

**Structure determination**. Three single-wavelength anomalous dispersion (SAD) datasets of the PP2A-TIPRL complex were processed and integrated using XDS[52]. Multiple processed datasets were scaled together by aimless[53]. Structures of TIPRL and PP2A–TIPRL were determined by SAD phasing using program CRANK[54] in the CCP4 package[55]. Selenium atoms were located by program AFRO/CRUNCH2[56]. Automatic model building and refinements for TIPRL and

PP2A–TIPRL were performed using AutoBuild[57] in PHENIX. The further model building was built manually using Coot[58] and refined using Phenix[59] with TLS restrains[60]. The structure of TIPRL was refined to 2.7 Å, and the free and working R-factors are 27.3 and 20.8%, respectively. The structure of PP2A–AC–TIPRL complex was refined to 3.8 Å, and the free and working R-factors are 24.6 and 19.7%, respectively. The crystallographic information is summarized in Table 1.

**Sequence alignment and conservation analysis.** Multiple sequence alignment was performed by CLUSTALW. The result was further submitted to ConSurf Server for calculating the conservation of residues of TIPRL among diverse species. The conservation scores are displayed by pymol as distinct colors using one copy of TIPRL in PP2A–TIPRL complex as the molecular model.

**Biolayer interferometry.** Biolayer interferometry (BLI) was used to measure the binding affinity between PP2A core enzyme and TIPRL similar to previously described[61]. In brief, BLI sensors with immobilized anti-GST antibody were activated by 150 nM GST-PP2A-AC dimer. Following wash, the activated sensors were dipped into increasing concentrations of TIPRL (0–30 μM) to measure the on-rate of TIPRL for 300 s and then dipped into the binding buffer to measure the off-rate for 300 s. Data collection and analysis were performed using ForteBio Octet RED96 and Data Analysis 9.0 (Pall Life Science), and the binding affinities were calculated based on the on- and off-rates by fitting to the 1:1 binding model.

**GST-mediated pulldown assay and competition in binding.** To examine the interaction between WT and mutant TIPRL with the methylated or non-methylated PP2A core enzyme in the presence or absence of indicated concentrations of OA or $Mn^{2+}$, 12 μg or titrated concentrations of TIPRL was mixed with 10 μl glutathione sepharose beads (Glutathione-Sepharose 4B (GS4B), GE Healthcare) with immobilized GST-tagged PP2A core enzyme (GST-Aα bound to Cα, 5–8 μg, with and without methylation) in a final volume of 100 μl assay buffer containing 25 mM Tris (pH 8.0), 150 mM NaCl, 2 mM dithiothreitol (DTT), supplemented with 1 mg ml$^{-1}$ bovine serum albumin to block nonspecific binding, and with and without OA or $Mn^{2+}$ as indicated. After incubation for 20 min, samples were washed three times with the assay buffer supplemented with 0.1% of Triton X-100, and the proteins bound to the resin were examined by sodium dodecyl sulfate-polyacrylamide gel electrophoresis (SDS-PAGE) and visualized by Coomassie blue staining. The methylation of the PP2A core enzyme was performed by mixing GST-AC dimer with a stoichiometric amount of LCMT-1, PTPA, and five stoichiometric amounts of S-adenosylmethionine in reaction buffer containing 25 mM Tris pH8.0, 150 mM NaCl, 20 mM DTT, and 50 μM $MnCl_2$. The reaction was carried out at 37 °C for 4 h followed by purification by Superdex 200 and examination of methylation level by western blot using an antibody that specifically recognizes the unmethylated PP2Ac (Millipore, 4b7, 1:1000).

The same method was also used to examine the interaction between TIPRL and the PP2A core enzyme bearing mutations in Aα or PP2Ac, in which the immobilized PP2A core enzyme contains mutations to residues at the interface to TIPRL. It is important to emphasize that we shortened the incubation time from 20 to 5 min to visualize the effects of mutations on the interaction between TIPRL and PP2A core enzyme for interaction assays. Some mutations still maintain the interaction with TIPRL if the incubation time is long enough, albeit much slower than that with WT PP2A core enzyme.

The test of these disease mutations found in cancer or patients with intellectual disability on holoenzyme assembly was performed similarly as above with regular incubation time (20 min). In brief, 12 μg of B'γ1 was mixed with GS4B resin with immobilized GST-tagged PP2A core enzyme, followed by the pulldown procedure described above. All experiments were repeated three times; representative results were shown. The uncropped images of SDS-PAGE mentioned above were shown in supplementary Figs. 12a–d and 13a–c.

**Thermal stability analysis.** Thermal shift assay was conducted with 1 mg ml$^{-1}$ of WT or mutant TIPRL in 50 mM MES buffer (PH 6.5), 150 mM NaCl, and 5× dilution of SYPRO Orange dye (Thermal Scientifics). Protein samples were heated with a 1 °C min$^{-1}$ increasing gradient from 20 to 90 °C, and the fluorescent intensity were detected with 1 min interval using the CFX Connect real-Time PCR detection system (Bio-Rad). Control assays were performed with buffer in the absence of a protein. Data were analyzed by fitting the fluorescence into Boltzmann model to obtain the midpoint temperature (accurate to zero digit) for the unfolding transition (Tm) of proteins as described before[62]. The experiments were performed three times. Tm from three independent experiments were averaged and mean±s.e. m. were calculated.

**GST-mediated competitive binding.** GST-mediated competitive binding followed a modified procedure similar to the pulldown assay described above. For TIPRL-mediated competitive inhibition of LCMT-1 or PTPA-binding to the PP2A core enzyme, 23 μg of LCMT-1 or PTPA was mixed with 10 μl GS4B resin with immobilized GST-tagged PP2A core enzyme in the presence or absence of titrated concentrations of TIPRL in a final volume of 250 μl assay buffer. For TIPRL-mediated competitive inhibition of binding of PP2A regulatory subunit, B'γ1, to the PP2A core enzyme, 6 μg of B'γ1 was premixed with 0, 10 and 100 μM of TIPRL in

50 μl binding buffer, followed by mixing with GS4B resin with immobilized GST-tagged PP2A core enzyme. The bound proteins were obtained and examined as described in the pulldown assay above. All experiments were repeated three times; representative results were shown. The uncropped images of SDS-PAGE are shown in supplementary Figs. 12e, f and 13d.

**Mammalian cell culture and western blot and co-immunoprecipitation.** Similar to what we previously described[12], the Flag-tagged human TIPRL (Flag-TIPRL), V5-tagged human PP2Ac α isoform (V5-PP2Ac), and their mutants were cloned into murine retroviral vectors bearing a cytomegalovirus promoter. After co-transfection into 293T cells (Thermal Scientific), the transfection and over-expression efficiency of Flag-TIPRL and V5-PP2Ac were monitored by western blot using antibodies that specifically recognize Flag-tag (Sigma, M2, 1:1000), V5-tag (Millipore, AB3792, 1:1000), or PP2Ac (Millipore, 1D6, 1:1000). The interaction between WT or mutant Flag-TIPRL and WT V5-PP2Ac was determined by co-immunoprecipitation using anti-FLAG affinity gel (Sigma, A2220), and the level of Flag-TIPRL and the associated PP2Ac were determined by western blot using antibodies mentioned above. The same method was applied to study the interaction between Flag-TIPRL and WT or mutant V5-PP2Ac. The original images of western blot are shown in supplementary Fig. 14.

**Disassembly of PP2A-B'γ1 holoenzyme by α4 and TIPRL.** A final concentration of 200 μg ml$^{-1}$ of the PP2A-B'γ1 holoenzyme was mixed with a near stoichiometric amount of GST-α4 in the presence or absence of 2 mM PPi or a stoichiometric amount of TIPRL. The mixture was incubated at 37 °C in the assay buffer described as in the pulldown assay supplemented with 50 μM of $Mn^{2+}$ to prevent spurious loss of catalytic metal ions. At each indicated incubation time, 50 μl aliquot was removed from incubation and mixed with GS4B resin to a final volume of 100 μl under the condition as described in the pulldown assay. After three washes with the assay buffer, the proteins bound to the resin were examined by SDS-PAGE and visualized by Coomassie blue staining. The experiments were repeated three times for multiple different batches of purified protein samples; representative results are shown. The uncropped images of SDS-PAGE are shown in supplementary Fig. 13e.

**Phosphatase assay.** PP2A phosphatase activity was measured using a phospho-Thr peptide as previously described[12]. Briefly, 2 μl of 1 mM phosphopeptide substrate (K-R-pT-I-R-R) was added to 60 nM methylated or unmethylated PP2A core enzyme or holoenzymes in a 20 μl reaction volume and in the presence and absence of titrated concentrations of TIPRL as indicated. After 15 min at 30 °C, the reaction was stopped by addition of 50 μl malachite green solution, and the absorbance at 620 nm was measured after 10 min incubation at room temperature. All experiments were performed in triplicate and repeated three times. Mean ± s.e.m. were calculated.

**Inactivation and dislodge of metal ions by α4 and TIPRL.** The active PP2A core enzyme (0.3 mg) was incubated at 37 °C alone, with TIPRL (0.3 mg), or with both α4 (0.3 mg) and TIPRL (0.3 mg) in the presence of 50 μM $Mn^{2+}$ in a reaction volume of 1.5 ml. At the indicated incubation time, a small aliquot was removed for monitoring the phosphatase activity using the phosphatase assay described above. At the end of 1-h incubation, free $Mn^{2+}$ was separated from PP2A by gel filtration chromatography. Metal contents of all three PP2A samples were then determined by ICP-MS.

**Data availability.** The atomic coordinates and structure factors have been deposited in the Protein Data Bank (www.pdb.org) with the accession numbers 5W0X and 5W0W for TIPRL (12-259 Δ94-103) and PP2A-TIPRL (12-259). Other data are available from the corresponding author upon reasonable request.

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

## Acknowledgements

We thank David Smith and Elena Kondrashkina (APS LS-CAT) for assistance on X-ray diffraction data collection, Drs Nathan Sherer (UW-Madison), Jeffrey Brodsky (University of Pittsburg), Emery Bresnick (UW-Madison), and Scott Horowitz (University of Denver) for reading the manuscript. The work is supported by R01 GM096060-01 (to Y.X.).

## Author contributions

A.Z. and C.-G.W. performed crystallographic studies, assisted by L.J., K.A.S. and V.S. Y. X., C.-G.W., A.Z. and K.A.S. determined the structures. C.-G.W., A.Z. and L.J. performed mutagenesis and biochemical studies, assisted by V.S., H.C., M.R., Y.L., B.J., T.-J.G. and Z.L. Y.X. guided all the studies and wrote the manuscript.

## Additional information

**Competing interests:** The authors declare no competing financial interests.

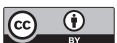

