## [Peer Review File · Nature Communications]

Reviewers' comments:

Reviewer #1 (Remarks to the Author):

This is an interesting and potentially important paper to understand a novel mechanism by which PP2A tumor suppressor is inactivated. By examining PP2A-inhibitory protein TIPRL alone and in complex with PP2A core enzyme, TIPRL interaction shows to be highly dependent on the methylation state of PP2Ac tail (L309) and is diminished in presence of L309 methyl moiety. The authors finally propose a model how PP2A holoenzyme disassembly and PP2Ac recycling could be achieved by coordinating activities of TIPRL and PP2A latency chaperone $\alpha 4$. The work presented would be the first evidential support of PP2A holoenzyme disassembly.

However there are number of issues that needs to further clarified and the conclusions need to be further supported by additional experimentation. Especially the authors need to demonstrate disassembly of a stable heterotrimeric holoenzyme by TIPRL.

Specific comments:

1. Lines 128/9: It is stated that (i) the inter-subunit PP2A-interactions are in the nanomolar range and that (ii) the binding affinity between TIPRL and core enzyme was measured at 0.3 μM (fig. 2f). In line 151, it is stated that TIPRL inhibited phosphatase activity of PP2A core enzyme with IC_{50} 0.3 μM (referring to Fig 2f). I don't see how could one get such values for IC_{50} from Fig. 2f Coomassie staining? From Fig. 2g, 50% AC phosphatase activity remained at TIPRL conc. $\sim 0.8\mu\text{M}$.

2. What is the stoichiometry of PP2A+TIPRL interaction?

3. What are the binding affinities PP2Ac (+/- methylation) and PP2A scaffolding subunit for TIPRL, and how do the values compare to the affinities for PR65+PP2Ac.

4. In lines 157-164, it is stated that (i) binding site of TIPRL on PP2Ac sterically prevents binding of PTPA and LCMT1 in the C-tail region of PP2Ac, and that (ii) TIPRL was also found to bind to holoenzymes containing non-methylated B56, but not PR70.

Are there any explanations for TIPRL-binding exclusion of PR70-containing PP2A holoenzymes? On Fig. 3b, on the overlaid structures please indicate TIPRL-binding region.

Is there any evidence (in assays involving recombinant proteins and in cells) to support the hypothesis that TIPRL binding would prevent LCMT1 and/ or PTPA binding to PP2A (PLA, competition assay...)?

5. In lines 179-182, it is stated that mutating residues at the TIPRL interface with scaffolding subunit abolishes interaction in vitro and in cells (Fig. 3c+3d). Is there evidence available how these mutations are tolerated by the protein (protein folding or stability measurements vs WT)?

For in cells statement (fig. 3d), where are the input samples and blot for PR65, as mutations are supposed to disrupt TIPRL-PR65? In Fig. 3f where is blot for TIPRL?

6. Fig.4c. Please measure phosphatase activity for AC alone (like positive ctrl) including and PPI + AC incubation as a control. representation for Mn^{2+} bound ions for AC alone. Error bars on Fig.4c (Mn^{2+} ions).

7. Quantifications for Fig.3e,f,g & Fig.4b. are needed

8. The most important potential novelty of the paper is the characterization of a mechanism for disassembly of a stable heterotrimeric holoenzyme. This aspect is also emphasized greatly by the

authors throughout the manuscript. Thereby, it is a major concern that in the materials and methods part, it is stated that TIPRL is pre-incubated with B56 and then added to GST-immobilised PP2A+C. What is the rationale for such outline? Why not PP2A heterotrimer + TIPRL, as the key hypothesis is TIPRL role in PP2A holoenzyme disassembly? Further, if binding assay in Fig. 3g was done with the same outline like described above (TIPRL + B56 pre-incubation), re-do this experiment so that TIPRL is titrated to PP2A heterotrimer. Additionally, demonstrate disassembly of the stable heterotrimer by gel filtration experiment. From line 187, disease-associated scaffolding mutations that have no effect on TIPRL binding. How do these mutations overlap with TIPRL-PR65 binding sites? How do these mutations affect PP2A heterotrimer assembly in cells (for different B56 families)?

Reviewer #2 (Remarks to the Author):

The authors report crystal structures of TIPRL alone and in complex with PP2A core enzyme. The complex structure shows that TIPRL binding causes the phosphatase active site perturbation, interfering with proper metal chelation. Although the complex structure is in relatively low resolution, comparison of 4 copies in the asymmetric unit reveals wobble contacts between TIPRL and scaffold subunit. This, together with biochemical studies, shows that disease mutations on PP2A/TIPRL interaction interface result in enhanced interaction with TIPRL while interfering regulatory subunit interactions, which are more specific. The reduction of active PP2A holoenzyme assembly and dysregulated cell signaling may explain the mechanism in a diseased state. Through structure guided studies, authors show that TIPRL binding to demethylated PP2A tail perturbs the active site, allowing $\alpha 4$ $\alpha 4$ binding and leading to PP2A holoenzyme disassembly, which is important for PP2Ac recycling. The role of TIPRL as a trigger of holoenzyme disassembly is critical to further our understanding on PP2A biogenesis and signaling. Therefore, I recommend this study for publication in Nature communication with minor revision on the following points.

1. Introduce components of heterotrimeric holoenzyme (i.e. catalytic, scaffold, regulatory) in the abstract.
2. Line 32, Include "latency chaperone, $\alpha 4$ ".... $\alpha 4$
3. Extended data 3d should be included in the main figure to help readers follow text.
4. Line 113, What is the RMSD among the 4 copies of complex, and also among each component?
5. Show Fig1 left in stereo.
6. Fig 2a stereo will be helpful
7. Fig 2d active PP2Ac side chain conformation needs to be shown.
8. Fig 2e, Metal chelation is not shown. This wire tracing of main chain is not informative.
9. Extended data 2 can be included in the main article.
10. Line 97, "Supplemental table 1" should be "Extended data Table 1".
11. Line 599, P305, the trajectory of R303 to T304 are not shown in cartoon.
12. Line 123, T138 not V138
13. Line 153, "(Figure 2g)" is missing.

Reviewer #3 (Remarks to the Author):

The manuscript by Xing and co-workers elucidates the structure of the enigmatic TIPRL protein in complex with PP2A AC core complex. The structure and accompanying analysis provides insights into how TIPRL might regulate the stability of PP2A holoenzymes. The insights into how carboxymethylation of the PP2Ac tail blocks interactions with TIPRL complement a structure of TIPRL recently published elsewhere. The biochemical tests of the interaction motifs support the authors contention that TIPRL may assist in dynamic removal (and perhaps interchange) of PP2A regulatory

subunits.

As a general comment - The literature on the biology of TIPRL is sparse, which is a bit surprising given the central role in PP2A regulation that is proposed in the author's model. While it is not the role of the authors to rectify this, it makes it harder to test predictions from their structural model in robust biological systems where alterations of TIPRL has been shown to have a phenotype.

Another issue to consider is the role of TIPRL in PP4 regulation, something reported in the literature but not mentioned here.

I have several minor issues.

Fig 2A tries to show several things at once. The overlay of the OA-bound PP2Ac is a bit confusing.

As a biochemist at heart, I am unfamiliar with the concept of "highly diverse modes" in crystallographic structures. A Google Scholar search for "wobble H-bond" in protein structures did not turn up much. Is this a novel concept, or is there precedent in the literature? If this is a new concept perhaps it could be stated more clearly.

Line 111: "More extensive and integrative contacts" What is an integrative contact? Could this statement 'more extensive' be quantitated?

Line 146: "ghost" metal ion. I don't understand what "ghost" means in this context. Perhaps this could be clarified.

In a number of assays the concentrations of the reactant is missing. For example figure 4b and 4C, please indicate the concentrations of all key reactants used.

The model in figure 5 is explained unclearly. There is an ungrammatical sentence describing 5b (lines 653-655) that adds to my confusion.

What is the intracellular concentration of TIPRL? Is there a phenotype to the knockout? I recognize these are not structural questions, but speak to the biological role of TIPRL.

Line 342: Argone National Laboratory has a typo: it is Argonne

We greatly appreciate that all 3 reviewers provided positive assessment of our manuscript. We thank them for their time and thoughtfulness in providing constructive comments that we have followed in making revisions. In response to the reviewer comments, we expanded our figure panels from 5 major figures to 7 major figures and 6 supplementary figures to 10 supplementary figures. The specific changes of figure numbers are not highlighted in the revised manuscript, but specifically mentioned in our response to reviewer comments. Our point-by-point responses are shown below, with reviewer comments in Arial and our response in Times New Roman. The corresponding changes are underlined in the revised manuscript.

Reviewer #1 (Remarks to the Author):

This is an interesting and potentially important paper to understand a novel mechanism by which PP2A tumor suppressor is inactivated. By examining PP2A-inhibitory protein TIPRL alone and in complex with PP2A core enzyme, TIPRL interaction shows to be highly dependent on the methylation state of PP2Ac tail (L309) and is diminished in presence of L309 methyl moiety. The authors finally propose a model how PP2A holoenzyme disassembly and PP2Ac recycling could be achieved by coordinating activities of TIPRL and PP2A latency chaperone $\alpha 4$. The work presented would be the first evidential support of PP2A holoenzyme disassembly.

However there are number of issues that needs to further clarified and the conclusions need to be further supported by additional experimentation. Especially the authors need to demonstrate disassembly of a stable heterotrimeric holoenzyme by TIPRL.

Specific comments:

1. Lines 128/9: It is stated that (i) the inter-subunit PP2A-interactions are in the nanomolar range and that (ii) the binding affinity between TIPRL and core enzyme was measured at 0.3 μM (fig. 2f). In line 151, it is stated that TIPRL inhibited phosphatase activity of PP2A core enzyme with IC_{50} 0.3 μM (referring to Fig 2f). I don't see how could one get such values for IC_{50} from Fig. 2f Coomassie staining? From Fig. 2g, 50% AC phosphatase activity remained at TIPRL conc. ~ 0.8 μM .

Response: Thanks for the reviewer to point out our miss of referring to Figure 2g for the IC_{50} of TIPRL in PP2A inhibition. Figure 2g was referred now on P7 of the revised manuscript: "TIPRL inhibited the phosphatase activity of the PP2A core enzyme with an IC_{50} of ~0.3 μM in the absence of free Mn^{2+} (Fig. 2g), comparable to the measured binding affinity between TIPRL and PP2A (Figure 2f)." For Figure 2g, the IC_{50} is 0.3 μM based on the calculation using Prism Graphpad software. The different conclusions from the reviewer and the software are most likely derived from two reasons: 1) the X axis of Figure 2g is plot in log scale, not linear, which might give an impression that IC_{50} is higher than calculated; 2) the software considers that the maximum inhibition for the assay results in a value of 10-20%, not zero. In this regard, the 50% inhibition would be 55-60% of the maximum activity. We consider the way the software handles the data is reasonable. To reinforce the conclusion, however, we also measured the binding affinity using bio-layer interferometry, which gave a value similar to the titration pulldown assay

and IC₅₀ shown above, and was added to the revised manuscript as Fig. 2f, right panel.

2. What is the stoichiometry of PP2A+TIPRL interaction?

Response: The PP2A-TIPRL complex purified via gel filtration chromatography showed that the interaction is 1:1 stoichiometry, which was further supported by the crystal structure. The titration pulldown assay shown in Figure 2f might indicate that TIPRL is sub-stoichiometric. This is most likely due to that the weak binding affinity between TIPRL and PP2A allow a portion of TIPRL fell off during wash.

3. What are the binding affinities PP2Ac (+/- methylation) and PP2A scaffolding subunit for TIPRL, and how do the values compare to the affinities for PR65+PP2Ac.

Response: The binding affinity of unmethylated PP2Ac and TIPRL was measured to be around 0.64 μM by titration pulldown assay (Supplementary Fig. 5a). Based on the pulldown assay between methylated PP2A-AC and TIPRL (Fig. 2f, central panel), there is no binding between methylated PP2Ac or PP2A core enzyme with TIPRL. Based on the pulldown assay and biolayer interferometry between unmethylated PP2A core enzyme and TIPRL, the measured binding affinity by both methods was 0.3μM (Fig.2f, left and right panels), which matches to the IC₅₀ measured from inhibition of PP2A core enzyme by TIPRL (Fig. 2g). The binding affinity between PP2A scaffold subunit and TIPRL was measured approximately 10 μM by titration pulldown assay (Supplementary Fig. 5b). We further tested the interaction between TIPRL and α4, and no interaction was detected. Since the data remains to be preliminary, we did not include it in the current manuscript. As a preliminary reference for the reviewer, the interactions among different components of PP2A subunits were summarized in the figure below.

4. In lines 157-164, it is stated that (i) binding site of TIPRL on PP2Ac sterically prevents binding of PTPA and LCMT1 in the C-tail region of PP2Ac, and that (ii) TIPRL was also found to bind to

holoenzymes containing non-methylated B56, but not PR70.

Are there any explanations for TIPRL-binding exclusion of PR70-containing PP2A holoenzymes? On Fig. 3b, on the overlaid structures please indicate TIPRL-binding region.

Is there any evidence (in assays involving recombinant proteins and in cells) to support the hypothesis that TIPRL binding would prevent LCMT1 and/or PTPA binding to PP2A (PLA, competition assay...)?

Response: The reason that the PP2A-PR70 holoenzyme might escape TIPRL attack even without methylation *in vitro* is likely because this holoenzyme is stabilized in the most compact conformation by the tripartite interactions of PR70 to the N-terminal edge and the top ridge of the scaffold subunit and to PP2Ac¹. The highly compact conformation might have left little room of “protein breathing” to allow TIPRL to access the PP2Ac-tail or the PP2A active site for the PP2A-PR70 holoenzyme.

Figure 3b (current 5b) has been modified to indicate the TIPRL-binding region on scaffold subunit.

We have performed the *in vitro* competition assay to examine the binding of immobilized PP2A core enzyme to PTPA or LCMT-1 in the presence of titrated concentrations of TIRPL. In response to this reviewer comments, we moved these data from supplementary figures (Supplementary Fig. 3b,c) to the main figures for the revised manuscript (Fig. 4b,c). The results showed TIPRL competes with PTPA and LCMT-1 for binding to the PP2A core enzyme in a concentration dependent manner, consistent with our structural observation that TIRPL binding overlaps with the binding sites of PTPA and LCMT-1 to the PP2A core enzyme.

Our studies in the past few years suggested that PP2A holoenzyme biogenesis follows a robust linear pathway, in which the latent form of PP2Ac is stabilized by $\alpha 4$ in a partially folded form²; upon activation by PTPA³, core enzyme formation and methylation would be drastically enhanced^{2,4} and drive toward holoenzyme assembly⁵. Although the cellular level of PTPA was reported to be highly abundant, the PP2A-PTPA complex could be barely detected. Although this interaction is capable of enhancing PP2A activation by Mg^{2+} by 10,000-fold³, the nature of the interaction is flimsy and transient and the majority of PP2A in cells is methylated and are present as holoenzymes^{6,7}. The dynamic molecular processes that control holoenzyme biogenesis or disassembly, albeit important, are highly challenging to decipher in cells. This notion also in part addresses the third reviewer’s comments, “The literature on the biology of TIPRL is sparse, which is a bit surprising given the central role in PP2A regulation that is proposed in the author’s model.”

To tackle such difficulties of investigating dynamic shuffling of PP2A complexes in cells, we are currently building a few lines of systematic research tools, including FRET (fluorescence resonance energy transfer) imaging, BiFC (biomolecular fluorescence complementation) imaging, co-IP, immunostaining, and split luciferase assay, etc. The PLA assay is also a great suggestion from this reviewer. We would be happy to include it as one of the approaches in our research given that the right antibodies are available. These tools, once built and validated, will be very valuable to track the dynamic changes of PP2A complexes in different signaling contexts. After these tools are built and validated, which will take a significant amount of time and effort,

we will be eager to tackle and demonstrate dynamic shuffling of PP2A complexes in cells under different signaling contexts in robust ways.

5. In lines 179-182, it is stated that mutating residues at the TIPRL interface with scaffolding subunit abolishes interaction *in vitro* and in cells (Fig. 3c+3d). Is there evidence available how these mutations are tolerated by the protein (protein folding or stability measurements vs WT)? For in cells statement (fig. 3d), where are the input samples and blot for PR65, as mutations are supposed to disrupt TIPRL-PR65? In Fig. 3f where is blot for TIPRL?

Response: We greatly appreciate the reviewer's meticulously careful examination of our data. TIPRL mutants used in the study were expressed in bacteria similar to WT TIPRL. In response to the reviewer comments, we did not observe changes of protein folding and expression level in mammalian cells. This was reflected by IP of Flag-TIPRL in Fig. 5d (original Fig. 3d). To corroborate this notion, we included the western blot data for recombinant Flag-TIPRL WT and mutants as well as the input for PP2Ac.

There are two reasons that we indirectly measure the interaction between PP2Ac and TIPRL, rather than the scaffold A-subunit. First, without PP2Ac, we could only detect a very weak interaction between A-subunit and TIPRL (Supplementary Fig. 5b). Without A-subunit, the interaction between PP2Ac and TIPRL is also weakened (Supplementary Fig. 5a) comparing to TIPRL-AC interaction (Fig. 2f, left and right panels). Second, the A-subunit is barely present alone in cells and is predominantly associated with PP2Ac. The rationale for detecting the interactions between PP2Ac and TIPRL to examine TIPRL mutations at the interface with the A-subunit is now added to P9 of the revised manuscript, "Consistent with our structural observations, in the absence of the scaffold subunit, the interaction between PP2Ac and TIPRL is weakened from 0.3 μ M (Fig. 2f, left and right panels) to \sim 0.6 μ M (Supplementary Fig. 5a), despite that only a very weak interaction could be detected between the scaffold subunit and TIPRL (Supplementary Fig. 5b). TIPRL mutations at the interface with the scaffold subunit weakened the interaction between TIPRL and PP2A both *in vitro* and in mammalian cells (Fig. 5c, d), underlying a novel finding that these contacts were important for PP2A-TIPRL interactions. Since PP2Ac is predominantly associated with the scaffold subunit in cells, disrupting the interactions between TIPRL and the scaffold subunit abolished the detection of interactions between TIPRL and PP2Ac (Fig. 5d)."

Thanks for the reviewer to identify the mistake in Figure 3f (Fig. 5f in the revised manuscript). The label in Figure 3f is now changed to "GST-PP2A-AC with A mutations pulldown B γ 1", rather than, "GST-PP2A-AC mutants pulldown WT TIPRL".

6. Fig.4c. Please measure phosphatase activity for AC alone (like positive ctrl) including and PPI + AC incubation as a control. representation for Mn²⁺ bound ions for AC alone. Error bars on Fig.4c (Mn²⁺ ions).

Response: AC control was included with limited data points for clarity. PPI evicts metal ions and completely inactivates the phosphatase activity within 15 minutes. This was reflected in our numerous previous studies. The relevant data can be found in two of our previous publications^{2,3}.

Since those experiments were not performed in the same experiments for $\alpha 4$ /TIPRL co-incubation with the PP2A core enzyme, these data were not included in the left figure panel of Supplementary Fig. 8 (original Fig. 4c). Although the data were collected in different experiments, PPI and $\alpha 4$ /TIPRL have clearly different kinetics in PP2A inactivation as the later took up to 1 hour to cause a near complete inactivation. We included PPI as a control to demonstrate that PP2Ac can be extracted from PP2A holoenzymes (Fig. 6b, original Fig. 4b) in part because the activity of PPI in this process was only reported once².

The level of Mn^{2+} ions associated with PP2A was detected by ICP-MS. This experiment is very costly to perform. Historically, this type of experiments was performed using single sample. Two of previous publications presented this data type as shown in original Fig. 4c right panel^{2,8}. In response to this reviewer comments, we moved original Fig. 4c to supplemental Fig. S8.

7. Quantifications for Fig.3e,f,g & Fig.4b. are needed

Response: The quantification results are now added to the figure panels suggested by the reviewer except Fig. 6b (original Fig. 4b). The quantification for Fig. 6b doesn't seem to improve the conclusion. The figure panel is a bit complex and additional information appears to make the figure more complicated. We hope the reviewer agrees with us.

8. The most important potential novelty of the paper is the characterization of a mechanism for disassembly of a stable heterotrimeric holoenzyme. This aspect is also emphasized greatly by the authors throughout the manuscript. Thereby, it is a major concern that the in the materials and methods part, it is stated that TIPRL is pre-incubated with B56 and then added to GST-immobilised PP2A+C. What is the rationale for such outline? Why not PP2A heterotrimer + TIPRL, as the key hypothesis is TIPRL role in PP2A holoenzyme disassembly? Further, if binding assay in Fig. 3g was done with the same outline like described above (TIPRL + B56 pre-incubation), re-do this experiment so that TIPRL is titrated to PP2A heterotrimer. Additionally, demonstrate disassembly of the stable heterotrimer by gel filtration experiment. From line 187, disease-associated scaffolding mutations that have no effect on TIPRL binding. How do these mutations overlap with TIPRL-PR65 binding sites? How do these mutations affect PP2A heterotrimer assembly in cells (for different B56 families)?

Response: We corrected the statement in the materials and methods, "TIPRL and B'γ1 is pre-mixed and then added to GST-AC immobilized on GS4B resin". The experiments in Figure 3g (Fig. 5g in the revised manuscript) and Figure 4b (Fig. 6b in the revised manuscript) are testing the role of TIPRL in two different molecular processes. Fig. 5g tested competitive TIPRL binding for blocking holoenzyme assembly, while Fig. 6d examined the ability of TIPRL and $\alpha 4$ in disassembly of PP2A holoenzymes. Our rationale is that, for the competitive binding, TIPRL and PP2A regulatory subunits co-exist in cells. For holoenzyme disassembly, however, the holoenzyme is the pre-existing form of PP2A that was under attack by $\alpha 4$ /TIPRL. Thus, in Fig. 6b, the experiments were performed using preformed holoenzymes, while in Fig. 5g, the experiments were performed by pre-mixing TIPRL and PP2A regulatory subunits.

To kindly remind the reviewer, the disease mutation residues at the TIPRL-A interface was illustrated in Supplementary Fig. 6. The test of these mutations on holoenzyme assembly in cells was published in a recent paper from a colleague in the field⁹, which was cited in the manuscript text on P10, “This result is in line with a recent observation that these mutations result in a higher of level TIPRL associated with cellular PP2A in cancer⁹.”

Reviewer #2 (Remarks to the Author):

The authors report crystal structures of TIPRL alone and in complex with PP2A core enzyme. The complex structure shows that TIPRL binding causes the phosphatase active site perturbation, interfering with proper metal chelation. Although the complex structure is in relatively low resolution, comparison of 4 copies in the asymmetric unit reveals wobble contacts between TIPRL and scaffold subunit. This, together with biochemical studies, shows that disease mutations on PP2A/TIPRL interaction interface result in enhanced interaction with TIPRL while interfering regulatory subunit interactions, which are more specific. The reduction of active PP2A holoenzyme assembly and dysregulated cell signaling may explain the mechanism in a diseased state. Through structure guided studies, authors show that TIPRL binding to demethylated PP2A tail perturbs the active site, allowing $\alpha 4$ binding and leading to PP2A holoenzyme disassembly, which is important for PP2Ac recycling. The role of TIPRL as a trigger of holoenzyme disassembly is critical to further our understanding on PP2A biogenesis and signaling. Therefore, I recommend this study for publication in Nature communication with minor revision on the following points.

1. Introduce components of heterotrimeric holoenzyme (i.e. catalytic, scaffold, regulatory) in the abstract.

Response: The description of PP2A heterotrimeric holoenzymes is now added to the revised manuscript on P3, “The broad substrate specificity and cellular functions of PP2A are controlled by the formation of diverse heterotrimeric holoenzymes; each contains a common core enzyme formed by the scaffold (A) and catalytic (C or PP2Ac) subunits and a third variable regulatory subunits from four major families (B/B55/PR55, B'/B56/PR61, B''/PR72, and B'''/Striatin)”.

2. Line 32, Include "latency chaperone, $\alpha 4$ ".... $\alpha 4$

Response: Thanks for the reviewer’s suggestion. “, $\alpha 4$ ” is now added to the abstract text.

3. Extended data 3d should be included in the main figure to help readers follow text.

Response: Thanks for the reviewer’s suggestion. The previous extended data Fig. 3 is now presented as Fig. 4.

4. Line 113, What is the RMSD among the 4 copies of complex, and also among each component?

Response: The RMSD among 4 copies of the complex, and among each component in different copies are shown in the table below. A brief description was added to the revised manuscript on P6: “the complex exhibits significant structural dynamics as reflected by its structural diversity of the four asymmetric copies in the crystal (Fig. 1), with the root-mean-square-deviation (RMSD) between different copies up to 2.6 Å, revealing dynamic structural fluctuation in TIPRL-PP2A interactions”, on P9, “The A-TIRPL wobble contacts might be in part contributed by the highly dynamic nature of the scaffold subunit (Supplementary Movie 1). Consistent with this notion, the RMSD of the scaffold subunit between different copies of the complex is up to 2.9 Å. In contrast, the RMSD of TIPRL between different copies is much smaller, 0.6-0.7 Å”, and on P11, “The RMSD of PP2Ac between different copies of the TIPRL-bound complexes is up to 1.4 Å, much higher than that of TIPRL.”

PP2A-A	A	D	G	J		TIPRL	B	E	H	K
A	-					B	-			
D	2.940	-				E	0.681	-		
G	2.222	1.530	-			H	0.649	0.686	-	
J	2.468	1.223	1.834	-		K	0.636	0.697	0.642	-
PP2Ac	C	F	I	L		Trimer	ABC	DEF	GHI	JKL
C	-					ABC	-			
F	0.741	-				DEF	2.556	-		
I	0.633	0.629	-			GHI	2.206	1.467	-	
L	1.291	1.386	1.363	-		JKL	2.476	1.334	1.869	-

* All values are in Angstroms

5. Show Fig1 left in stereo.

Response: The Fig1 left panel is now shown in stereo.

6. Fig 2a stereo will be helpful

Response: The Fig. 2a is now shown in stereo.

7. Fig. 2d active PP2Ac side chain conformation needs to be shown.

Response: The side chain conformations of active PP2Ac are now shown in Fig. 2d.

8. Fig 2e, Metal chelation is not shown. This wire tracing of main chain is not informative.

Response: A figure inlet is now added to Fig. 2e to show the side chain conformations of active PP2Ac and the TIPRL-bound PP2Ac

9. Extended data 2 can be included in the main article.

Response: Previous extended data 2 is now shown as Fig. 3.

10. Line 97, “Supplemental table 1” should be “Extended data Table 1”.

Response: “Supplemental table 1” is now good for the format required for *Nature Communication*. Our original manuscript was initially prepared for *Nature* and got transferred to *Nature Communication*.

11. Line 599, P305, the trajectory of R303 to T304 are not shown in cartoon.

Response: We showed the trajectory of main chains of R303 to T304 in cartoon at right panel (since the electronic density is not good enough for us to determine the position of their side chains). Since the description might be confusing, we removed the following text - “P305 to L309 of PP2Ac are shown in sticks and the trajectory of R303 to T304 are shown in cartoon.” from the corresponding figure legend for Fig. 2b.

12. Line 123, T138 not V138

Response: We greatly appreciate that the reviewer identified this error. It is now fixed in the revised manuscript.

13. Line 153, “(Figure 2g)” is missing.

Response: The missing information is now added to the revised manuscript.

Reviewer #3 (Remarks to the Author):

The manuscript by Xing and co-workers elucidates the structure of the enigmatic TIPRL protein in complex with PP2A AC core complex. The structure and accompanying analysis provides insights into how TIPRL might regulate the stability of PP2A holoenzymes. The insights into how carboxymethylation of the PP2Ac tail blocks interactions with TIPRL complement a structure of TIPRL recently published elsewhere. The biochemical tests of the interaction motifs support the authors contention that TIPRL may assist in dynamic removal (and perhaps interchange) of PP2A regulatory subunits.

1. As a general comment - The literature on the biology of TIPRL is sparse, which is a bit surprising given the central role in PP2A regulation that is proposed in the author’s model. While it is not the role of the authors to rectify this, it makes it harder to test predictions from their structural model in robust biological systems where alterations of TIPRL has been shown to have a phenotype.

Response: There could be many reasons why TIPRL literature is sparse. We attempted to provide three reasons here: 1) PP2A regulation is highly complex. The multifaceted biochemical mechanisms of TIPRL might make it difficult to decipher its mechanisms in mammalian cells; 2)

Although TIPRL interaction with PP2A is expected to be important for PP2A regulation, the resulting complex represents a minor species of PP2A complexes in mammalian cells, and thus did not draw enough attention to the field and our research community; 3) Since TIPRL is expected to regulate dynamic shuffling of PP2A holoenzymes, the phenotype caused by alteration of TIPRL might be related to different PP2A holoenzymes in different signaling contexts, which might greatly complicate initial understanding of TIPRL function in mammalian cells.

The structural and biochemical dissection of PP2A regulation provide a certain level of advantages to tackle complex PP2A regulation network. Our effort on this aspect in the past few years suggested that PP2A holoenzyme biogenesis follows a robust linear pathway, in which the latent form of PP2Ac is stabilized by $\alpha 4$ in a partially folded form²; upon activation by PTPA³, core enzyme formation and methylation would be drastically enhanced^{2,4} and drive toward holoenzyme assembly⁵. The molecular events en route of holoenzyme biogenesis have similar nature of interactions with PP2A as TIPRL. These dynamic molecular events are crucial but difficult to study in mammalian cells because the majority of PP2A in cells are methylated and are predominantly present as holoenzymes⁴. One interesting example is PTPA. While the cellular level of PTPA is twice as high as PP2A, but the PP2A-PTPA complex could be barely detected. Although this interaction is capable of enhancing PP2A activation by Mg^{2+} by 10,000-fold³, the flimsy nature of the interaction makes it impossible to decipher the mechanism and function in mammalian cells.

Holoenzyme biogenesis and recycling provides appealing mechanisms for dynamic changes of PP2A holoenzymes for broad cellular signaling and processes. The structural insights we obtained will enable us to tackle difficulties of investigating dynamic shuffling of PP2A complexes in cells. Guided by structural information, we are currently building a few lines of research tools, including FRET (fluorescence resonance energy transfer) imaging, BiFC (bimolecular fluorescence complementation) imaging, co-IP, immunostaining, and split luciferase assay, etc. We are hoping that these tools, once built and validated, will be valuable to track dynamic changes of PP2A complexes in different signaling contexts and alleviate difficulties of investigating PP2A regulation in vivo. This will also allow us to examine how changes of PP2A holoenzymes might be affected by TIPRL in different signaling contexts.

2. Another issue to consider is the role of TIPRL in PP4 regulation, something reported in the literature but not mentioned here.

Response: Our reviewer made a very important point here. PP4 and PP6 are considered PP2A-like phosphatases; they have a much higher sequence similarity to PP2A than other members of the PPP family. Consistently, PP4 and PP6 shared several regulatory proteins, including PP2A methylation enzyme, latency chaperone $\alpha 4$. Indeed, TIPRL was also shown to interact with PP4 and PP6. To predict whether the mode of interaction between PP4 and TIPRL is similar to PP2A-TIPRL, we performed sequence alignment of PPP family serine/threonine phosphatases with residues involved in TIPRL interaction highlighted above the sequence (Supplementary Fig. 10). Indeed, all the residues participate in TIPRL binding are identical between PP2A and PP4, and only one residue is different in PP6, suggesting that TIPRL interacts with PP4 and PP6 in a

similar mode. Interestingly, the helix switch near the TIPRL binding site has identical sequence in PP4 and PP6, underlying that the forces created by TIPRL binding might lead to similar structural strain in PP4 and PP6 to trigger its dissociation from holoenzymes. These notions are now added to the DISCUSSION of the revised manuscript on P15, “It is important to mention that PP4 and PP6 are considered PP2A-like phosphatases; they have a much higher sequence similarity to PP2A than other members of the PPP family.⁵Consistently, PP4 and PP6 shared several regulatory proteins with PP2A, including methylation enzyme, latency chaperone $\alpha 4^2$, and inhibitory protein TIPRL¹⁰. TIPRL was also found to suppress PP4 activity during DNA damage response¹¹. To predict whether the mode of interaction between PP4/PP6 and TIPRL is similar to PP2A-TIPRL, we performed sequence alignment of PPP family serine/threonine phosphatases to examine residues involved in TIPRL interaction (Supplementary Fig. 10). All the residues in PP2A that participate in TIPRL binding are identical in PP4, and only one residue is different in PP6, suggesting that TIPRL interacts with PP4 and PP6 in a similar mode as with PP2A. Interestingly, the helix switch near the TIPRL binding site has identical sequence in PP4 and PP6 as well, suggesting that the forces created by TIPRL binding might lead to similar structural strain in PP4 and PP6 to trigger their dissociation from holoenzymes”.

Minor issues:

1. Fig 2A tries to show several things at once. The overlay of the OA-bound PP2Ac is a bit confusing.

Response: The overlay of OA-bound PP2Ac is now shown separately in Supplementary Fig. 2.

2. As a biochemist at heart, I am unfamiliar with the concept of “highly diverse modes” in crystallographic structures. A Google Scholar search for “wobble H-bond” in protein structures did not turn up much. Is this a novel concept, or is there precedent in the literature? If this is a new concept perhaps it could be stated more clearly.

Response: In response to this reviewer comment, we presented the close-up views of the contacts between TIPRL and the PP2A scaffold subunit in all four copies of the complex (Supplementary Fig. 4), and explained on P9 of the revised manuscript on what “wobble contacts” refer to, “The tolerance of different modes of interactions at the periphery of the A-TIPRL interface resembles the tolerance of different nucleotides at the third wobble position of codons. We thus refer to these contacts as “wobble contacts”.”

3. Line 111: “More extensive and integrative contacts” What is an integrative contact? Could this statement ‘more extensive’ be quantitated?

Response: The text is revised as the reviewer suggested.

4. Line 146: “ghost” metal ion. I don’t understand what “ghost” means in this context. Perhaps this could be clarified.

Response: In response to this reviewer comment, the text is revised as “with a misplaced metal ion left several angstroms from the active site that would not confer any enzymatic activity (Fig. 2e).” on P7 of the revised manuscript.

5. In a number of assays the concentrations of the reactant is missing. For example figure 4b and 4C, please indicate the concentrations of all key reactants used.

Response: The experiments for Fig. 6b had been performed many times by at least three lab members, using at least 6 different batches of purified proteins. This was stressed now in the Methods section of the revised manuscript on P21, “The experiments were repeated three times for multiple different batches of purified protein samples; representative results were shown”. As we described in Methods, TIPRL concentrations were slightly over 1 μ M, similar to the holoenzyme concentration. The visual inspection of the 10 μ l input shown on Fig. 4b (now Fig. 6b) suggested that the TIPRL concentration is close to 2 μ M, and might be higher for the methylated PP2A holoenzyme. This falls in the range of difference in protein concentration estimation by different methods. In brief, the experiments were performed in a low μ M range. We apologize that we missed the reaction volume for Fig. 4c (now Supplementary Fig. 8), which was now added to the Methods “in a reaction volume of 1.5 ml”. This volume was chosen for convenience of sample injection for gel filtration chromatography.

6. The model in figure 5 is explained unclearly. There is an ungrammatical sentence describing 5b (lines 653-655) that adds to my confusion.

Response: In response to this reviewer comment, we revised the figure legend as follows, “**Fig. 7. Cartoon models for α 4/TIPRL-mediated PP2A holoenzyme disassembly and recycling of PP2Ac into the latent PP2Ac- α 4-TIPRL complex.** (a) Cartoon illustration of mechanisms protecting methylated holoenzyme from TIPRL/ α 4 attack. In the methylated holoenzyme, the methylated PP2Ac tail hinders TIPRL-binding, and thus weakens TIPRL’s ability to attack the phosphatase active site. The intact active site would hinder α 4-binding and subsequent global conformational changes¹². The cartoon for the PP2A scaffold subunit is colored green, PP2Ac blue, regulatory subunit brown, TIPRL magenta, α 4 yellow, and helix and loop switches of PP2Ac red. Catalytic metal ions are indicated by red dots. (b) Cartoon Illustration of TIPRL/ α -mediated disassembly of unmethylated holoenzyme. Without methylation, the holoenzyme is subjected to initial attacks by TIPRL, which induces partial dislodge of catalytic metal ions and potential dissociation of the regulatory subunit, as well as partial perturbation of helix and loop switches near the PP2A active site to allow α 4 docking. The combined force of α 4 and TIPRL triggers global conformational switches of PP2Ac that completely release catalytic metal ions, and expel both scaffold and regulatory subunits, resulting in latent PP2Ac- α 4-TIPRL complex.”

7. What is the intracellular concentration of TIPRL? Is there a phenotype to the knockout? I recognize these are not structural questions, but speak to the biological role of TIPRL.

Response: We are hoping to address the predicted biological function of TIPRL in the near future as we respond to the first major comment of this reviewer. The cellular concentration of TIPRL had not been reported yet. One of our future plan is to examine lysates from different

mouse tissues by western blot and use purified TIPRL with titrated concentrations as standard to measure the cellular concentration of TIPRL. We expect that the cellular concentration would vary in different tissues and in response to different cellular and environmental cues. For example, PP2A holoenzymes were suggested to fall apart during DNA damage response⁶. And TIPRL was found to suppress PP2A holoenzyme activity ATM/ATR DNA damage signaling¹⁰. These points were discussed on P13, “It is also intriguing to consider that periodic demethylation of PP2A occurs during cell cycle¹², so that PP2A holoenzymes may undergo cell cycle-dependent disassembly by $\alpha 4$ and TIPRL and their phosphatase activity may fluctuate similar to cell cycle-dependent kinases, but in opposite phases. Moreover, $\alpha 4$ /TIPRL-mediated recycling might also contribute to down-regulation of PP2A holoenzymes during stress or DNA damage responses⁶. TIPRL was found to suppress PP2A activity in DNA damage-induced ATM/ATR signaling¹⁰”

We do not expect that the cellular concentration of TIPRL could be as high as 1 μ M, a concentration we chose for the convenient of *in vitro* biochemical assays. Among many of our repeats for Fig. 6b, some experiments were performed with lower concentrations of TIPRL (when the available amount of PP2A holoenzyme was low). Similar results were observed. It is important to note that the cellular PP2A holoenzymes are nearly 100% methylated, albeit *in vitro*, methylation does not affect holoenzyme assembly, suggesting that the cellular concentration of TIPRL is sufficient for surveillance of PP2A holoenzyme integrity. This point was also discussed on P13, “TIPRL’s attack of the PP2A active site requires that the PP2Ac tail be demethylated (Fig. 2), which reconciles a long-standing controversy regarding the role of carboxymethylation on PP2A holoenzyme assembly *in vitro* and *in vivo*. Methylation was found to be essential for holoenzyme assembly in cells^{13,14} but not *in vitro*^{15,16}”.

8. Line 342: Argone National Laboratory has a typo: it is Argonne

Response: Thanks for the reviewer to identify the typo. It is now corrected in the revised manuscript.

References:

1. Wlodarchak, N., *et al.* Structure of the Ca(2+)-dependent PP2A heterotrimer and insights into Cdc6 dephosphorylation. *Cell Res* **23**, 931-946 (2013).
2. Jiang, L., *et al.* Structural basis of protein phosphatase 2A stable latency. *Nat Commun* **4**, 1699 (2013).
3. Guo, F., *et al.* Structural basis of PP2A activation by PTPA, an ATP-dependent activation chaperone. *Cell Res* **24**, 190-203 (2014).
4. Stanevich, V., *et al.* Mechanisms of the scaffold subunit in facilitating protein phosphatase 2A methylation. *PLoS One* **9**, e86955 (2014).
5. Wlodarchak, N. & Xing, Y. PP2A as a master regulator of the cell cycle. *Crit Rev Biochem Mol Biol* **51**, 162-184 (2016).
6. Kong, M., Ditsworth, D., Lindsten, T. & Thompson, C.B. $\alpha 4$ is an essential regulator of PP2A phosphatase activity. *Mol Cell* **36**, 51-60 (2009).
7. Stanevich, V., *et al.* The Structural Basis for Tight Control of PP2A Methylation and Function by LCMT-1. *Mol Cell* **41**, 331-342 (2011).

8. Xing, Y., *et al.* Structural mechanism of demethylation and inactivation of protein phosphatase 2A. *Cell* **133**, 154-163 (2008).
9. Haesen, D., *et al.* Recurrent PPP2R1A Mutations in Uterine Cancer Act through a Dominant-Negative Mechanism to Promote Malignant Cell Growth. *Cancer Res* **76**, 5719-5731 (2016).
10. McConnell, J.L., Gomez, R.J., McCorvey, L.R., Law, B.K. & Wadzinski, B.E. Identification of a PP2A-interacting protein that functions as a negative regulator of phosphatase activity in the ATM/ATR signaling pathway. *Oncogene* **26**, 6021-6030 (2007).
11. Rosales, K.R., *et al.* TIPRL Inhibits Protein Phosphatase 4 Activity and Promotes H2AX Phosphorylation in the DNA Damage Response. *PLoS One* **10**, e0145938 (2015).
12. Longin, S., *et al.* Spatial control of protein phosphatase 2A (de)methylation. *Exp Cell Res* **314**, 68-81 (2008).
13. Tolstykh, T., Lee, J., Vafai, S. & Stock, J.B. Carboxyl methylation regulates phosphoprotein phosphatase 2A by controlling the association of regulatory B subunits. *Embo J* **19**, 5682-5691 (2000).
14. Ogris, E., Gibson, D.M. & Pallas, D.C. Protein phosphatase 2A subunit assembly: the catalytic subunit carboxy terminus is important for binding cellular B subunit but not polyomavirus middle tumor antigen. *Oncogene* **15**, 911-917 (1997).
15. Xu, Y., *et al.* Structure of the protein phosphatase 2A holoenzyme. *Cell* **127**, 1239-1251 (2006).
16. Xing, Y., *et al.* Structure of protein phosphatase 2A core enzyme bound to tumor-inducing toxins. *Cell* **127**, 341-353 (2006).

Reviewers' comments:

Reviewer #1 (Remarks to the Author):

The authors have clearly improved the manuscript and we appreciate very comprehensive responses to our question. In particular, the figure drawn to explain interactions among PP2A components is very nice and I could suggest that even to be included as supplementary figure for the paper. However the following points need to be addressed before the far-reaching conclusions of the paper are properly supported by the data.

Specific comments:

1. In response to our point 5, the authors do not provide any convincing evidence how well these mutations are tolerated by the protein. They claim that that proteins were expressed in bacteria similar to WT but this do not address the question of protein quality directly. As this is a critical experiment for conclusions, it is essential that authors provide direct measurement data of folding properties of the mutant recombinant proteins as compared to WT by using for example thermal shift analysis. Authors further claim that mutant proteins would be similarly expressed as WT proteins in mammalian cells (as an indication of proper folding) but the WB shown in Fig. 5d does not allow such conclusions as it is very heavily overexposed and thus masks clearly lower expression of Mut proteins that is readily evident even from this overexposed blot. Authors need to provide data with blots that are exposed only to level that is shown in panel 5f and to provide quantitation from three independent experiments with standard deviation.

2. The most important potential novelty of the paper is the characterization of a mechanism for disassembly of a stable heterotrimeric holoenzyme. This aspect is also emphasized greatly by the authors throughout the manuscript. The authors have provided sufficient responses to most of our criticism of the data directly assessing this question in the previous round, but considering the importance of the data shown in current Fig. 6 for the concept, the authors have not yet sufficiently scrutinized the finding.

Quantifications for current Fig.5e,f,g and 6b are needed.

-Any quantitation needs to be shown as an average of at least 3 independent experiments +/- standard deviation. This is a very basic quality requirement for any data presented in this level journal. Showing quantitation of one arbitrary selected WB does provide any information about how repeatable and robust the phenotype is

-It is almost ridiculous to claim that "The quantification for Fig. 6b doesn't seem to improve the conclusion. The figure panel is a bit complex and additional information appears to make the figure more complicated". Without quantifications and information whether the presented data are from the same experiment/gels, the comparison between different conditions is impossible. Again, an average of 3 experiments +/- standard deviations are needed for the efficiency of PP2Ac release from the complex (using input in the same gel as a control). Further, the most important comparison, capacity of a4 alone vs. a4+TIPRL to release PP2Ac from the trimer has to be presented from samples run on the same gel to allow direct comparison of efficiency

Minor points:

1. The title gives an impression that the principle of ".....Decommissioning of Multimeric Complexes" would have been validated with several different complexes. As this is not the case, I would strongly recommend modifying the title as "".....Decommissioning of Multimeric PP2A Complex"

2. Please mark IC50 values to the Fig. 2g as it makes it much easier to correlate to data in Fig. 2f.

We greatly appreciate that our reviewers agreed with most of the points of our responses to previous comments, and that reviewer 1 provided more insights and advices for further improvement of the manuscript. Our point-by-point responses are shown below, with reviewer comments in Arial and our response in Times New Roman. The corresponding changes are underlined in the revised manuscript

Reviewer #1 (Remarks to the Author):

Overall comment: The authors have clearly improved the manuscript and we appreciate very comprehensive responses to our question. In particular, the figure drawn to explain interactions among PP2A components is very nice and I could suggest that even to be included as supplementary figure for the paper. However the following points need to be addressed before the far-reaching conclusions of the paper are properly supported by the data.

Response: We greatly appreciate that the reviewer found the summary figure for the reviewers on the interactions among PP2A components useful. We agree with the reviewer on the importance of this follow-up study. The data as outlined now is still preliminary though. We are in the process to address the gap in the summary figure and plan to publish completed results in a separate paper.

Specific comments:

Comment #1. In response to our point 5, the authors do not provide any convincing evidence how well these mutations are tolerated by the protein. They claim that that proteins were expressed in bacteria similar to WT but this do not address the question of protein quality directly. As this is a critical experiment for conclusions, it is essential that authors provide direct measurement data of folding properties of the mutant recombinant proteins as compared to WT by using for example thermal shift analysis.

Response: To confirm that TIPRL mutations used in our mutagenesis studies do not affect the folding and thermal stability of TIPRL, we have performed the thermal shift assay for comparing the thermal stability of WT and mutant TIPRL that we generated to examine important residues involved interactions with PP2A-A or PP2Ac. Experimental methods and data of thermal shift assay are now added to the “Method” session and a supplementary figure 2. A brief description of the results is added to P7 of the revised manuscript, “Mutation of TIPRL residues at either interface abolished or weakened interactions between TIPRL and the PP2A core enzyme (Fig. 3d) without obvious alteration of the thermal stability of TIRPL

(Supplementary Fig. 2).”

Specifically, the results showed that most of mutants have similar T_m as WT besides that E159R and R184E have $\sim 4^{\circ}$ C higher and L182R has $\sim 6^{\circ}$ C lower T_m than WT. The reasons why E159R and R184E have higher thermal stability than WT are intriguing (Fig. 2b and Fig. 5a upper right). It is possibly that side chain of R159 interacted with sidechain of E157 to stabilize the structure which is more repulsive without mutation. The enhanced thermal stability for the R184E mutant, on the other hand, remains to be further studied. Although the thermal stability of L182R are much less, based on the result of pulldown assay, the interaction between TIPRL L182R and PP2Ac is still sustained. The reason why this mutation affected the stability remains to be investigated. One plausible explanation is that the L182R mutation disturbs the hydrophobic interaction between side chain of L182 with nearby F180 and/or a repulsive force is generated toward to the sidechain of R184 (Fig. 2b) that destabilizes the local structural packing.

Authors further claim that mutant proteins would be similarly expressed as WT proteins in mammalian cells (as an indication of proper folding) but the WB shown in Fig. 5d does not allow such conclusions as it is very heavily overexposed and thus masks clearly lower expression of Mut proteins that is readily evident even from this overexposed blot. Authors need to provide data with blots that are exposed only to level that is shown in panel 5f and to provide quantitation from three independent experiments with standard deviation.

Response: We have replaced the overexposed blot with the data obtained by shorter exposure time (Fig. 5f).

Comment #2. The most important potential novelty of the paper is the characterization of a mechanism for disassembly of a stable heterotrimeric holoenzyme. This aspect is also emphasized greatly by the authors throughout the manuscript. The authors have provided sufficient responses to most of our criticism of the data directly assessing this question in the previous round, but considering the importance of the data shown in current Fig. 6 for the concept, the authors have not yet sufficiently scrutinized the finding.

Quantifications for current Fig.5e, f, g and 6b are needed.

-Any quantitation needs to be shown as an average of at least 3 independent experiments +/- standard deviation. This is a very basic quality requirement for any data presented in this level journal. Showing quantitation of one arbitrary selected WB does provide any information about how repeatable and robust the phenotype is
-It is almost ridiculous to claim that “The quantification for Fig. 6b doesn’t seem to improve the conclusion. The figure panel is a bit complex and additional information

appears to make the figure more complicated". Without quantifications and information whether the presented data are from the same experiment/gels, the comparison between different conditions is impossible. Again, an average of 3 experiments +/- standard deviations are needed for the efficiency of PP2Ac release from the complex (using input in the same gel as a control). Further, the most important comparison, capacity of a4 alone vs. a4+TIPRL to release PP2Ac from the trimer has to be presented from samples run on the same gel to allow direct comparison of efficiency

Response: Our previous publications on structural biology did not provide the level of quantification and statistical analysis for data that extend our structural insights. Nonetheless, the reviewer made a valid point. In response to this reviewer comments, quantifications for Figs.5e, and 5f with standard deviation calculated from 3 independent experiments are now added to these figure panels.

For Fig.5g, we illustrated the quantified results by concentration-dependent curves for better clarity instead of using a set of numbers. Value for each point on the graph was calculated from 3 independent experiments with error bars (standard deviation). For Fig.6b, we illustrated the quantified results by time-dependent curves to show the efficiency of the recycling; the results are presented in a new panel (Fig. 6c). Value for each point on the graph was calculated from 3 independent experiments with error bars.

The reason we didn't run the results of recycling PP2Ac from the holoenzyme by a4 alone vs. a4+TIPRL on same SDS-PAGE in Fig.6b is because there weren't enough lanes in a single gel to run all the samples simultaneously.

Minor points:

1. The title gives an impression that the principle of ".....Decommissioning of Multimeric Complexes" would have been validated with several different complexes. As this is not the case, I would strongly recommend modifying the title as "".....Decommissioning of Multimeric PP2A Complex"

Response: Thanks for the review's suggestion. We agreed with this comment and have modified the title as "Methylation-Regulated Decommissioning of Multimeric PP2A Complexes".

2. Please mark IC50 values to the Fig. 2g as it makes it much easier to correlate to data in Fig. 2f.

Response: The IC50s are now shown on Fig2g.

REVIEWERS' COMMENTS:

Reviewer #1 (Remarks to the Author):

The authors have now properly addressed all the criticism.